# Digenic inheritance of mutations in *EPHA2* and *SLC26A4* in Pendred syndrome

Mengnan Li[1,2], Shin-ya Nishio[3], Chie Naruse[4], Meghan Riddell[1], Sabrina Sapski[5], Tatsuya Katsuno[6], Takao Hikita[1], Fatemeh Mizapourshafiyi[1,2], Fiona M. Smith[7], Leanne T. Cooper[7], Min Goo Lee [8], Masahide Asano[4], Thomas Boettger [9], Marcus Krueger [10], Astrid Wietelmann[11], Johannes Graumann[12,13], Bryan W. Day[7], Andrew W. Boyd[7], Stefan Offermanns[5], Shin-ichiro Kitajiri [3], Shin-ichi Usami[3] & Masanori Nakayama [1,2,14✉]

Enlarged vestibular aqueduct (EVA) is one of the most commonly identified inner ear malformations in hearing loss patients including Pendred syndrome. While biallelic mutations of the *SLC26A4* gene, encoding pendrin, causes non-syndromic hearing loss with EVA or Pendred syndrome, a considerable number of patients appear to carry mono-allelic mutation. This suggests faulty pendrin regulatory machinery results in hearing loss. Here we identify *EPHA2* as another causative gene of Pendred syndrome with *SLC26A4*. EphA2 forms a protein complex with pendrin controlling pendrin localization, which is disrupted in some pathogenic forms of pendrin. Moreover, point mutations leading to amino acid substitution in the *EPHA2* gene are identified from patients bearing mono-allelic mutation of *SLC26A4*. Ephrin-B2 binds to EphA2 triggering internalization with pendrin inducing EphA2 autophosphorylation weakly. The identified EphA2 mutants attenuate ephrin-B2- but not ephrin-A1-induced EphA2 internalization with pendrin. Our results uncover an unexpected role of the Eph/ephrin system in epithelial function.

[1] Laboratory for Cell Polarity and Organogenesis, Max Planck Institute for Heart and Lung Research, Bad Nauheim, Germany. [2] DFG Research Training Group, Membrane Plasticity in Tissue Development and Remodeling, GRK 2213, Philipps-Universität Marburg, Marburg, Germany. [3] Department of Otorhinolaryngology, Shinshu University School of Medicine, Matsumoto, Japan. [4] Institute of Laboratory Animals, Graduate School of Medicine, Kyoto University, Kyoto, Japan. [5] Department of Pharmacology, Max Planck Institute for Heart and Lung Research, Bad Nauheim, Germany. [6] Department of Otolaryngology - Head and Neck Surgery Kyoto University Graduate School of Medicine, Kyoto, Japan. [7] QIMR Berghofer Medical Research Institute, Brisbane, Australia. [8] Department of Pharmacology, Yonsei University College of Medicine, Seoul, Korea. [9] Department of Cardiac Development and Remodelling, Max Planck Institute for Heart and Lung Research, Bad Nauheim, Germany. [10] Institute for Genetics and Cologne Excellence Cluster on Cellular Stress Responses in Aging-Associated Diseases (CECAD), University of Cologne, Cologne, Germany. [11] MRI and µCT Service Group, Max Planck Institute for Heart and Lung Research, Bad Nauheim, Germany. [12] Scientific Service Group Biomolecular Mass Spectrometry Max Planck Institute for Heart and Lung Research, Bad Nauheim, Germany. [13] German Centre for Cardiovascular Research (DZHK), Partner Site - Rhine-Main, Berlin, Germany. [14] Kumamoto University International Research Center for Medical Scinece, Kumamoto, Japan. ✉email: masanori.nakayama@mpi-bn.mpg.de

The inner ear comprises a fluid-filled epithelial tube within a spiral space of the temporal bone. Inner ear malformation, such as enlarged vestibular aqueduct (EVA), is a characteristic feature of Pendred syndrome, an autosomal recessive disorder defined by hearing loss and goitre. Pendrin, also known as sodium-independent chloride/iodide transporter, is an anion exchanger protein that in humans is encoded by the *SLC26A4* gene (solute carrier family 26, member 4)[1–4]. Pendrin was identified as a causative gene of Pendred syndrome[1] and non-syndromic hearing loss with EVA[5]. Various mutations in the *SLC26A4* gene associated with Pendred syndrome and non-syndromic hearing loss with EVA have been identified[6]. While Pendred syndrome patients do not exhibit serious symptom in the kidney[4], pendrin is predominantly expressed in the inner ear, thyroid and kidney[4]. Some reported mutations on *SLC26A4* cause an impaired pendrin transporter function, others affect protein folding, localization, or trafficking of pendrin[6]. Roughly 50–80% of patients with EVA, have mutations in *SLC26A4*[7–9]. Genetic inactivation of *Slc26a4* in mice established a causative role of pendrin on inner ear dysfunction[10]. Although Pendred syndrome is autosomal recessive, mono-allelic mutations of *SLC26A4* are often found in patients who are diagnosed with Pendred syndrome or non-syndromic hearing loss with EVA. Moreover, some patients diagnosed with Pendred syndrome do not carry any mutations of the *SLC26A4* gene[9]. These observations suggest that compromised pendrin regulatory machinery may result in hearing loss, however, the regulator of pendrin remains elusive.

Eph receptors constitute the largest family of receptor tyrosine kinases (RTKs) and interact with plasma-membrane-bound ephrin ligands. Ephrins are divided into two subclasses, A subclass (ephrin-A1-A5 in mammals), which are tethered to the cell membrane by GPI-anchor, and B subclass (ephrin-B1-B3), which are characterized by a transmembrane domain, followed by a short cytoplasmic domain. Ephs are divided on the basis of sequence similarity and ligand affinity into A (EphA1-A8 and A10) and B (EphB1-B4 and B6) subclasses[11]. In the nervous system, Ephs are known to be involved in axon guidance, neural crest cell migration, compartment boundary formation and synapse formation[12–15]. Ephs and their ligands also play important roles in vascular development[16,17]. We have previously shown that ephrin-B2 and EphBs binding regulate vascular morphogenesis. EphB/ephrin-B2 regulates vascular endothelial growth factor receptor (VEGFR) 2 and 3 internalization in endothelial cell[18,19], suggesting functional roles of Eph/ephrins in transmembrane protein localization. Eph/ephrin interactions also govern the compartmentalization of cells in epithelial tissue formation. Loss of ephrin-B2 in the epithelium results in abnormal morphology of the inner ear[20].

Here we identify *EPHA2* as another causative gene of Pendred syndrome with *SLC26A4*. Pendrin is identified as a binding partner of EphA2. *EphA2* knock out (KO) mice exhibit abnormal structures in epithelial tissues with mislocalization of pendrin, including in the inner ear and thyroid. Given the phenotypic similarity of inner ear epithelial ephrin-B2 deficient mice with pendrin loss of function mice[20], we re-examine the binding of ephrin-B2 with EphA2. Although it is assumed that the ligands of class-A Eph receptors are generally ephrin-As, we find that ephrin-B2 forms a complex with EphA2. Stimulation of EphA2 with ephrin-B2 induces internalization of EphA2 and pendrin from the plasma membrane leading to weaker autophosphorylation of EphA2 than that with ephrin-A1. We find that a subset of mutated forms of pendrin identified from Pendred syndrome patients do not bind to EphA2 and is not regulated after stimulation of EphA2 with ephrin-B2. Moreover, we identify EphA2 mutations in patients diagnosed with non-syndromic hearing loss with EVA carrying mono-allelic mutation of *SLC26A4*. Induction of the identified EphA2 mutations reveals a compromised binding ability of EphA2

with ephrin-B2, but not ephrin-A1. As a result, ephrin-B2 induced pendrin internalization is impaired. Our result reveals an unexpected relationship between the Eph/ephrin system and Pendred syndrome.

## Results

### Identification of pendrin as a binding partner of EphA2.
EphA2 is highly expressed in epithelial cells in vitro and in vivo[21,22]. In the kidney, immunoreactive signal against an EphA2 antibody was confirmed in the lectin Dolichos biflorus agglutinin (DBA) positive collecting duct (Supplementary Fig. 1a). Strong expression of EphA2 in the basement membrane of the kidney epithelium in the collecting duct was observed, which was diminished in kidney sections from *EphA2* knock out mice (Supplementary Fig. 1b, c)[23]. Interestingly, a specific signal corresponding to EphA2 was also seen in dot-like structures at the apical region of the epithelial cells (Supplementary Fig. 1c). To gain further insight into the role of EphA2 in epithelial cells, we carried out an interactome analysis of EphA2. Immunoprecipitation of EphA2 from the mouse kidney tissue using a specific antibody against EphA2 was examined. Western blotting analysis has shown that 80% of total EphA2 in the lysate was precipitated (Fig. 1a). After confirmation of specific precipitation by silver staining (Fig. 1b), the eluted proteins were analyzed by liquid chromatography with tandem mass spectrometry (LC-MS/MS), compared to an IgG control using label free quantification. Given the important role of Eph/ephrin system as a regulator of co-receptors[18,19,24,25], we focused on transmembrane proteins for further analysis. Among them, *Slc26a4* (pendrin) protein was identified with the highest enrichment (Fig. 1c). To confirm the interaction of EphA2 and pendrin, a tagged version of both proteins was overexpressed in HEK293T cells. When myc-tagged pendrin was immunoprecipitated with an anti-myc-tag-antibody, V5-EphA2 was co-precipitated (Supplementary Fig. 1d). Moreover, the intracellular domain of EphA2 with the transmembrane domain was co-precipitated with myc-tagged pendrin but not a construct containing the extracellular domain of EphA2 with the transmembrane domain (Supplementary Fig. 1e). In addition to its expression in the kidney, pendrin is known to be expressed at the apical surface of the cells in the spiral prominence of the cochlear duct in the inner ear and in the follicular cells in the thyroid[26–28]. Pendrin was co-immunoprecipitated with EphA2 from both the inner ear and thyroid lysates as well as from kidney lysates of control mice but not those of *EphA2* KO mice (Fig. 1d). Interestingly, immunoreactive signal against EphA2 antibody was restricted to the apical surface of the cells at the spiral prominence of the cochlear duct in the inner ear (Fig. 1e), while it was observed at both the apical and basal membrane of the follicular cells in the thyroid (Fig. 1f).

### EphA2 KO mice show pendrin mis-localization.
Given the fact that EphA2 was identified as a binding partner of pendrin, the phenotype of *EphA2* KO mice was reanalyzed in the inner ear and the thyroid. Although overt structural changes, such as those seen in *Slc26a4* KO mice[10], were not observed in the overall *EphA2* KO mice cochlea with micro-CT scan, the size of the cochlear duct was significantly enlarged (Fig. 2a, b). H&E staining of the inner ear revealed a decreased thickness of intermediate cell layer of the stria vascularis in *EphA2* KO mice, which consisted of three layers of cells; marginal, intermediate, and basal cells (Fig. 2c–e). Additionally, we found *EphA2* KO mice showed obvious goitre and thyroid hypertrophy (Supplementary Fig. 2a, b). H&E staining of thyroid glands further confirmed an enlarged follicle lumen area (Fig. 2f, g).

Next, we carried out immunostaining with an anti-pendrin antibody in the inner ear, the thyroid and the kidney. Pendrin

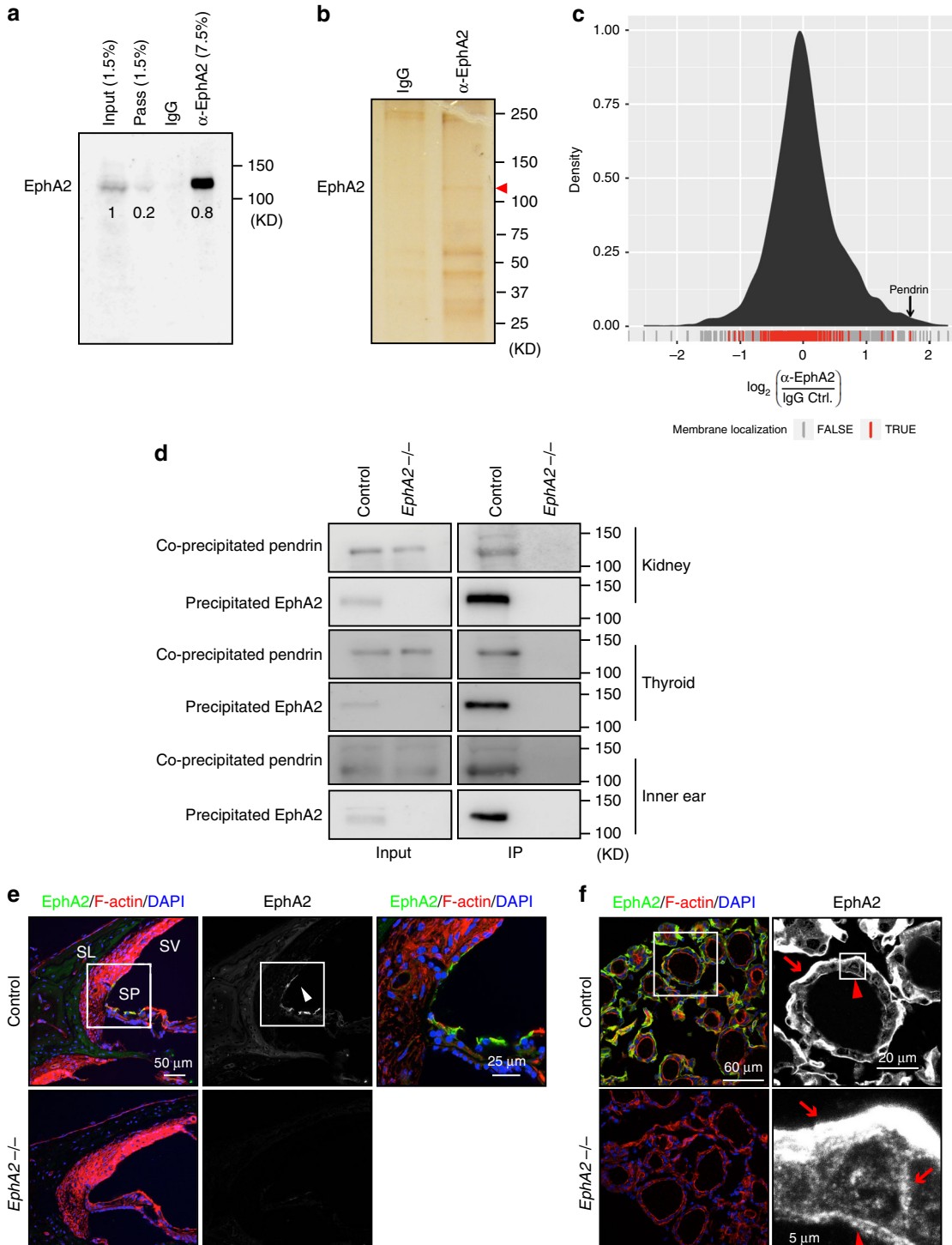

**Fig. 1 Identification of pendrin as an interacting protein of EphA2. a, b** Immunoprecipitation of EphA2 from kidney lysate. Specific precipitation of EphA2 was confirmed by western blotting analysis (**a**) and by silver staining (**b**) of precipitated protein compared to IgG control. Indicated numbers in **a** represent relative amount of precipitated EphA2 and EphA2 passed thorough immunoprecipitation compared to input (pass). **c** Identification of pendrin by label free quantification in mass spectrometry-based protomics analysis of EphA2 immunoprecipitate as compared to IgG control. Pendrin is the protein with the highest enrichment among membrane localized proteins. The ratio distribution of proteins identified is shown as a kernel density estimate. Localization of transmembrane proteins in the distribution is indicated red in the rug. **d** Confirmation of the protein complex by immunoprecipitation using an anti-EphA2 antibody. Precipitated proteins from the kidney, the thyroid and the inner ear from control and *EphA2* KO mice were analyzed by western blotting analysis with the EphA2 and the anti-pendrin antibodies respectively. **e** Immunostaining of the inner ear using anti-EphA2 specific antibody (green or gray, arrowhead) with phalloidin (red, F-actin) and DAPI (blue, nucleus). The specificity of the signal against EphA2 was confirmed in the *EphA2* KO inner ear. SL spiral ligament, SV stria vascularis, SP spiral prominence. **f** Immunostaining of the thyroid using anti-EphA2 specific antibody (green) with phalloidin (red, F-actin) and DAPI (blue, nucleus). The specific signal of EphA2 was confirmed in the *EphA2* KO thyroid. The strong signal of EphA2 was observed at basal membrane (arrow). Although the signal is weak, EphA2 expression was also confirmed at apical surface (arrowhead). Source data are provided as a Source Data file.

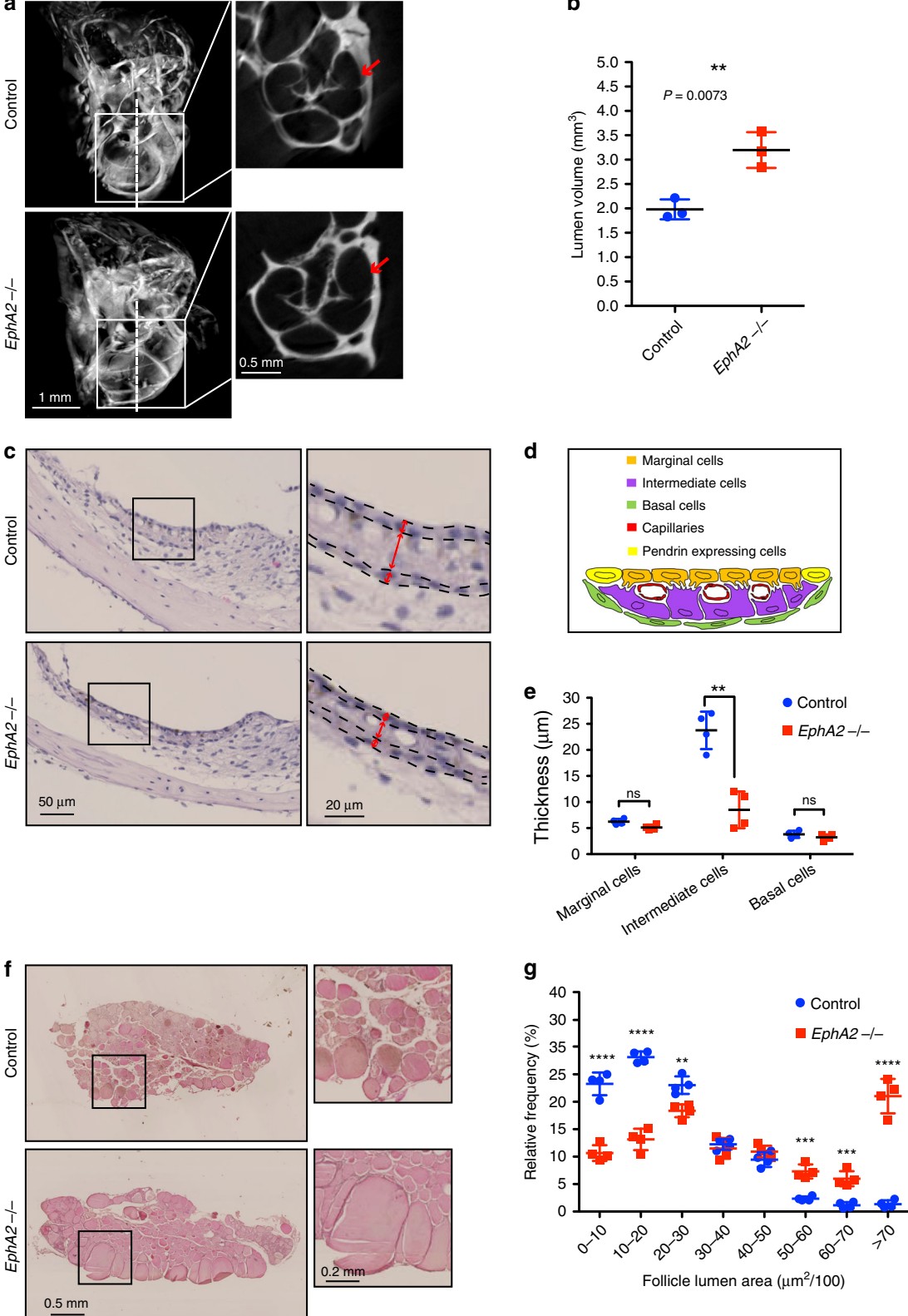

**Fig. 2 Morphological analysis of *EphA2* KO mice phenotype in the inner ear and thyroid. a** View of the inner ear of control (upper panels) and *EphA2* KO mice (lower panels) visualized by μCT. Arrows in the sagittal section indicate the lumen in the cochlea. **b** Lumen volume was quantified. Lines indicate mean value from three animals with SD. ***p < 0.01 by *t*-test. **c**–**e** Thickness of stria vascularis in the *EphA2* KO mice was reduced compared to that in control mice. **c** H&E staining of the inner ear are shown. Dot-line indicates the boarder of cell layers. **d** SV was consisted of mainly three cell types, maginal cells, intermedia cells and basal cells as shown in (d). H&E staining of the inner ear revealed a decreased thickness of the intermedia cell layter in *EphA2* KO mice. **e** Average data from 4 animals are presented. Lines indicate mean value from four animals with SD. ***p < 0.01 by student *t*-test. **f**, **g** *EphA2* KO mice exhibited Thyroid goitre. H&E staining of thyroid sections are shown in (**f**). Proportion of follicle lumen area in control and EphA2 KO mice is presented in the graph (**g**). Mean ± SD.; one-way ANOVA; ****p < 0.0001, ***p < 0.001, **p < 0.01. Source data are provided as a Source Data file.

expression was restricted to the apical surface of epithelial cells of the spiral prominence in the cochlea in control mice, meanwhile it was delocalized from the apical surface and observed as dot-like structures in *EphA2* KO mice (Fig. 3a). Moreover, mis-localization of pendrin from the apical surface to the intracellular region was also observed in follicle cells in the thyroid of *EphA2* KO mice (Fig. 3b). In the kidney, pendrin localization was observed on the apical surface of epithelial cells in the renal tube (Fig. 3c). Conversely, pendrin was found in an epithelial-epithelial junctional pattern and intracellular vesicle like structures in the *EphA2* KO mice as well as disrupted apical surface staining (Fig. 3c). These results indicate that EphA2 controls pendrin localization in vivo. As the loss of pendrin expression in the inner ear is known to lead to reduced expression of KCNJ10 protein, presumably causing deafness[26], KCNJ10 expression in stria vascularis was examined in *EphA2* KO mice. The signal corresponding to KCNJ10 was reduced in *EphA2* KO mice (Fig. 3d, e), suggesting a functional relationship between EphA2 and pendrin.

**Ephrin-B2 is a ligand of EphA2.** A recent study has shown that inner ear epithelial cell specific *Efnb2*, the gene encoding ephrin-B2, deficient mice exhibit vestibular-behavioral dysfunction and signs of abnormal endolymph homeostasis, which is often seen in Pendred syndrome patients[29]. Ephrin-B2 is required for the growth and morphogenesis of the endolymphatic sac and duct of the inner ear[20]. Although the EphA2 receptors are known to be activated by ephrin-A but not ephrin-B ligands[30], we re-examined the relationship between ephrin-B2 and EphA2. While the ligands of EphAs are assumed to be ephrin-As, EphA4 has been shown to bind to ephrin-B2. The crystal structure of the EphA4 ligand-binding domain in complex with ephrin-B2 has already been resolved (Fig. 4a)[31]. When the structure of EphA2 was superimposed to EphA4 in the complex with ephrin-B2, we hypothesized that EphA2 could possibly bind to ephrin-B2 (Fig. 4a). Prompted by this prediction, we immunoprecipitated ephrin-B2 from mouse kidney lysate and confirmed that ephrin-B2 was co-precipitated with EphA2 from kidney lysate of control mice but not that of *EphA2* KO mice (Fig. 4b). Next, we incubated soluble ephrin-B2 extracellular domain-Fc fusion protein with MDCK II cell lysate and found that endogenous EphA2 was precipitated with the extracellular domain of ephrin-B2 (Fig. 4c). To confirm the interaction of EphA2 with ephrin-B2, we calculated the Kd value between extracellular domains of the ephrin-B2 and the EphA2. The fc-fusion protein of the EphA2 extracellular domain was immobilized onto the ELISA plate and binding of that of ephrin-B2 was examined. The Kd value between ephrin-B2 and EphA2 was 85 nM (Supplementary Fig. 3a), while that between EphA2 and ephrin-A5, the classic ligand of EphA2 was 2 nM (Supplementary Fig. 3b). The binding of ephrin-B2 with EphA4 as a ligand has also been analyzed with a previous observation by Bowden et al. determining the Kd values of EphA4/ephrin-B2, EphA4/ephrin-A4, and EphA4/ephrin-A5, respectively[32]. The Kd value of EphA4/ephrin-B2 was 10.8 μM, which ~30 times greater than that of EphA4/ephrin-A5 (Kd = 360 nM) and 300 times greater than that of EphA4/ephrin-A4 (Kd = 36 nM). Thus, the Kd value of EphA2/ephrin-B2 was within a reasonable range (Supplementary Fig. 3a, b)[32].

The stimulation of RTKs by their ligand proteins induces receptor dimerization and auto-phosphorylation[33]. Consistent with previous findings, stimulation of MDCK II cells expressing EphA2 with ephrin-A1 extracellular domain FC-fusion protein, the classical ligand of EphA2, promoted auto-phosphorylation of EphA2 at Tyr594 in 15 min, which was sustained at least up to 2 h (Fig. 4d, e). The level of EphA2 auto-phosphorylation induced by ephrin-B2

was weaker than that by ephrin-A1, but observed in 15 min after stimulation (Fig. 4d, e). Under these conditions, treatment of the cells with human Fc failed to induce EphA2 auto-phosphorylation (Fig. 4d, e).

After stimulation with ligands, RTKs are internalized and transported to early endosome[33]. Both ephrin-A1 and ephrin-B2 triggered internalization of cell surface EphA2 in a time dependent manner (Fig. 4f, g). Interestingly, cell surface pendrin was also internalized upon both ephrin-A1 and ephrin-B2 stimulation (Fig. 4f, h). Consistent with the in vivo observations (Fig. 3a–c), knock down of EphA2 reduced surface presentation of pendrin. Under the condition, cell surface pendrin was not internalizad by ephrin-B2 (Supplementary Fig 3c, d). Furthermore, we observed co-localized immunoreactive signal corresponding to ephrin-B2, phospho-EphA2 and EEA1, a marker of early endosomes, 2 h after ephrin-B2 stimulation (Supplementary Fig. 3e, f). In addition, we have confirmed that immunoreactive signal corresponding to the anti-ephrin-B2 antibody was colocalized with that to the anti-EphA2 antibody in the inner ear (Supplementary Fig. 3g). These results suggest an important role of ephrin-B2 as an inducer of EphA2 endocytosis with the transmembrane binding partner, pendrin, while its effect is weaker than that of ephrin-A1.

**Aberrant regulation of pathogenic forms of pendrin via EphA2.** Several amino-acid substitutions of pendrin have been identified from Pendred syndrome patients as well as non-syndromic hearing loss patients with EVA. To gain further insight into the relationship between EphA2 and pendrin, we examined the interaction of pathogenic forms of pendrin with EphA2. Among reported mutations, introduction of some identified mutations including pendrin Pro123 to Ser (P123S), Val 138 to Phe (V138F), Leu236 to Pro (L236P), Ala372 to Val (A372V), Asn392 to Tyr (N392Y), Thr416 to Pro (T416P), Leu445 to Trp (L445W), Gln446 to Arg (Q446R), Gly497 to Ser (G497S), Val653 to Ala (V653A), Ser666 to Phe (S666F), Gly672 to Glu (G672E), Thr721 to Met (T721M), and His723 to Arg (H723R) have been found to be delocalized from the plasma-membrane to the cytosol in cultured cells (Supplementary Fig. 4)[6,34–38]. When these pathogenic forms of pendrin were transfected and immunoprecipitated with the EphA2 cytoplasmic domain containing trans-membrane domain, myc-pendrin A372V, L445W, Q446R, and G672E were not co-immunoprecipitated with EphA2 (Fig. 5a, b).

To gain further insight into the role of EphA2 on pendrin regulation, pendrin A372V, L445W, Q446R or G672E was co-overexpressed with EphA2. The cells were transfected with cDNAs of encoding myc-pendrin diease forms with that of EphA2, and the non-permeable cells were stained with an anti-myc antibody. While signal corresponding to myc-pendrin was observed in ~65% of cells, ratio of V5-pendrin A372V, L445W, Q446R, or G672E positive cells was significantly decreased (Supplementary Fig. 5a, b). Under these conditions, co-expression of EphA2 did not affect protein expression levels of these pathogenic forms of pendrin (Fig. 5a) but partially restored membrane localization of myc-pendrin A372V, L445W or Q446R (Supplementary Fig. 5a, b). On the other hand, EphA2 overexpression did not affect localization of G672E.

The substitutions of Leu117 to Phe (L117F), Ser166 to Asn (S166N), and Phe335 to Leu (F335L), identified in Pendred syndrome patients, do not affect their membrane localization[6,36]. Given the reported normal function of pendrin L117F and pendrin S166N as an anion exchanger[6,36], compromised regula-tory machinery of pendrin function may cause the observed symptoms. To examine whether EphA2 is involved in dysfunction of pendrin caused by these amino acid substitutions, the effect of pendrin L117F, pendrin S166N, and pendrin F355L mutations on EphA2 interaction and internalization was examined. While the

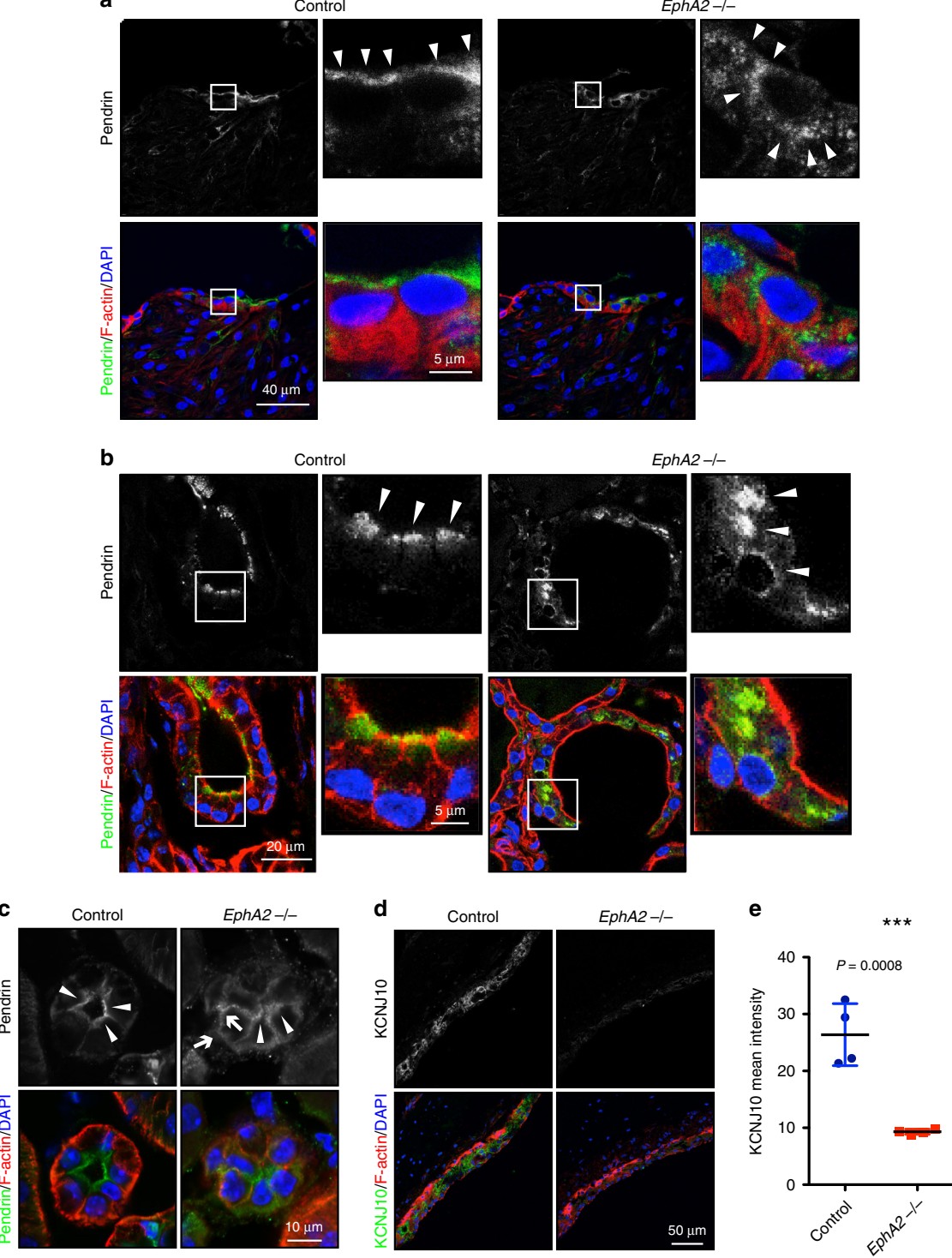

**Fig. 3 Compromised localization of pendrin in *EphA2* KO mice. a** Immunostaining of pendrin in the inner ear of control and *EphA2* KO mice. Green, pendrin; red, actin; blue, DAPI. Arrowheads indicate the immunoreactive signal corresponding to pendrin. **b** Immunostaining of pendrin in the thyroid of control and *EphA2* KO mice. Green, pendrin; red, actin; blue, DAPI. Arrowheads indicate the immunoreactive signal corresponding to pendrin. **c** Immunostaining of pendrin in the kidney of control and *EphA2* KO mice. Green, pendrin; red, actin; blue, DAPI. Arrows and arrowheads indicate the immunoreactive signal corresponding to pendrin. **d** Immunostaining of KCNJ10 in the inner ear of control and *EphA2* KO mice. KCNJ10 expression in stria vascularis shown in green was reduced in *EphA2* KO mice. **e** Average data from four animals are presented. Lines indicate mean value from 4 animals with SD. ***$p < 0.001$ by Student's $t$-test. Source data are provided as a Source Data file.

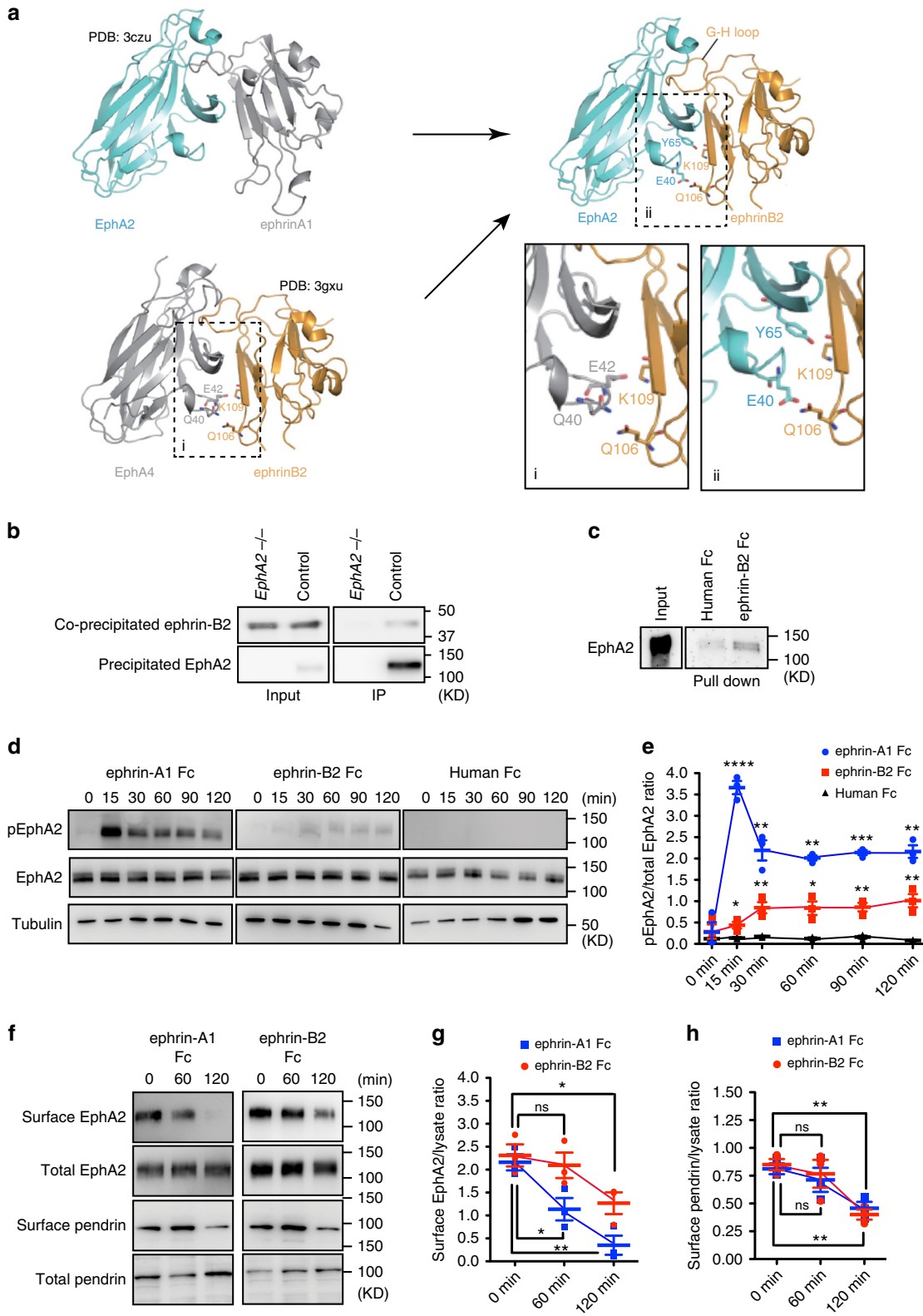

amount of co-precipitated pendrin mutants with EphA2 was comparable to that of wild type (wt) pendrin (Fig. 5c, d), the S166N mutant failed to be internalized after ephrin-B2 stimulation (Fig. 5e, f). Taken together, these results further demonstrate that EphA2 could control both pendrin recruitment to the plasma membrane and pendrin exclusion from the plasma membrane.

**EPHA2 mutations in pendred syndrome patients**. Among sensorineural hearing loss patients EVA or the Pendred syndrome patients, a considerable number of patients carry one copy of the mutation on the *SLC26A4* gene, therefore a compromised pendrin regulatory machinery may be involved in the pathogenesis of the syndrome. To further analyze the role of EphA2 in Pendred

**Fig. 4 Ephrin-B2 is a ligand of EphA2. a** Overall structure of the EphA2/ephrin-A1, EphA4/ephrin-B2, and EphA2/ephrin-B2 complexes. Protein Data Bank codes are indicated. In addition to a loose fit of the ephrin-B2 G-H loop in the EphA4 ligand-binding channel, a second contact region in the EphA4-ephrin-B2 interface involving extensive surface polar contacts may form binding of EphA4 to ephrin-B2 (i). The side chain hydrogen bond between Gln40 of EphA4 and Gln106 of ephrin-B2 (ii) and the side chain salt bridge between Glu42 and Lys109 of ephrin-B2 (i) might form the binding of ephrin-B2 and EphA4. **b** Immunoprecipitation of ephrin-B2 with EphA2. Precipitated proteins analyzed by immunoblot. **c** Pull-down analysis using ephrin-B2-Fc fusion protein. Co-precipated EphA2 is shown by immunoblot analysis. **d** The effect of ephrin-B2 and ephrin-A1 stimulation on EphA2 autophosphorylation. Cultured MDCK II cells were stimulated with ligands for the indicated time. **e** Relative amount of phospho-EphA2 is shown. Mean ± SEM.; one-way ANOVA; ****$p < 0.0001$, ***$p < 0.001$, **$p < 0.01$, *$p < 0.05$; ($n = 3$). **f** The effect of ephrin-B2 and ephrin-A1 stimulation on EphA2 and pendrin internalization. Samples were analyzed by western blotting with indicated antibodies. **e** Relative amount of cell surface EphA2 (**g**) or pendrin (**h**) is shown. Mean ± SEM.; one-way ANOVA; *$p < 0.05$; ($n = 3$). Source data are provided as a Source Data file.

syndrome, direct sequencing of the *EPHA2* gene in 40 Japanese hearing loss patients with EVA carrying mono-allelic mutation of *SLC26A4* were examined. While mutation of ~70 genes causing hearing loss were previously identified as a human nonsyndromic deafness gene[39], they were not identified in these patients[40]. On the other hand, two missense mutations of the *EPHA2* gene were identified in two families, *SLC26A4*: c.1300G>A (p.434A>T), *EPHA2*: c.1063G>A (p.G355R) and *SLC26A4*: c.1229C>A (p.410T>M), *EPHA2*: c.1532C>T (p.T511M) (Fig. 6a, b). These *EPHA2* mutations were predicted to be pathological by several in silico prediction software programs (Supplementary Table 1). The patient carrying c.1300G>A of *SLC26A4* was previously reported[41]. She is 22-years-old and presented congenital bilateral sensorineural hearing loss, goitre and skin disorders, while her brother, sister and parents were healthy and did not show such symptoms (Fig. 6a). Another patient carrying c.1229C>A of *SLC26A4* is 14-years-old and exhibited progressive, symmetrical sensorineural hearing loss and goitre (Fig. 6c). Her mother was healthy and carries a mutation of the *EPHA2* gene but not the *SLC26A4* gene (Fig. 6b). A temporal bone CT scan of the patient revealed an EVA (Fig. 6d). Previously, a haplotype of 12 uncommon variants upstream of the *SLC26A* gene was identified to presumably cause the syndrome in patients carrying one coding mutation of the *SLC26A* gene in the United States and Denmark, while the EVA-associated haplotype (CEVA) is rare in Asian polulations[42,43]. Consistently, these variants were not identified from these two patients.

**Mutated EphA2 affects its ligand-binding specificity.** The ligand binding domain (LBD), Cysteine-rich domain (CRD) and fibronectin 1 (FN1) domain of EphA fold into a rigid, rod-like structure with little interdomain flexibility. Conversely, the connection between FN1 and FN2 is flexible. Interestingly, the EphA2 mutations identified in the patients, Gly355 and Thr511, are located in proximity of the linker region between Sushi/EGF domain and FN1, and between the FN1 and FN2, respectively (Supplementary Fig. 6a)[44]. A crystal structure of ephrin-A5 and EphA4, which has high sequence homology with EphA2, has been reported[45]. An extracellular domain of EphA2 with EphA4 with identified mutations is presented (Supplementary Fig. 6a). Interestingly the sites of the identified EphA2 mutations are conserved among all the other human EphA and EphB. Moreover, Gly355 is conserved in *C. elegans* Eph, while Thr511 is conserved in Drosophila Eph, Dek (Supplementary Fig. 6b). These facts suggest an essential role of these amino acids on Eph function that might be conserved throughout evolution. To examine whether these mutations affect the ligand-binding specificity of EphA2 to ephrin-A and ephrin-B, a pull down assay was performed with HEK293T cells due to their low level of endogenous EphA2 expression (Supplementary Fig 7a, b)[15]. While tagged versions of EphA2 G355R and EphA2 T511M were effectively precipitated with Fc-fusion ephrin-A1 compared to EphA2 WT, Fc-fusion ephrin-B2 failed to pull down EphA2

G355R and T511M (Fig. 7a). Consistently, internalization of EphA2 G355R and EphA2 T511M with pendrin induced by ephrin-B2 but not ephrin-A1 was suppressed (Fig. 7b, c). On the other hand, the mutated forms of EphA2 did not affect their ability to bind to pendrin (Fig. 7d).

**Discussion**

Proper and polarized localization of transporters in cells is essential for their function. Various previously identified pendrin mutations cause pendrin cytoplasmic localization. A subset of these mutations, such as H723R, are known to cause mis-folding of the protein, leading to accumulation in the endoplasmic reticulum[35,46]. Low temperature incubation and salicylate treatment of cultured cells, which are thought to help with protein-folding processes, rescues the membrane localization of H723R[35]. On the other hand, mis-localization of pendrin A372V from the plasma membrane is not restored by these treatments[35], suggesting these mutations may affect pendrin trafficking from the Golgi to the plasma membrane but not protein-folding. Here, we found that pendrin A372V, L445W, Q446R, and G672E did not bind to EphA2. Given the fact that loss of EphA2 disturbs pendrin apical localization in vivo and cell surface presentation in vitro, the binding of pendrin with EphA2 might be critical for pendrin recruitment to the apical membrane in the inner ear and the thyroid. Thus, loss of the ability of pendrin to bind EphA2 may cause delocalization of pendrin from the plasma membrane. Additionally, we examined the binding ability of EphA2 to four membrane located forms of mutated pendrin. None of the mutants had impaired interaction with EphA2. However, S166N, which is known to have an intact transporter activity and membrane localization in cultured cells[36], showed compromised endocytosis after ephrin-B2 stimulation. Multiple types of ephrin-A are expressed in the inner ear during development[47], meanwhile inner ear epithelial cell specific *Efnb2* KO mice exhibit EVA-like malformation[20]. While the function of ephrin-A5 in the inner ear was investigated[48], no reports have shown that any ephrin-A single KO mice show inner ear malformation. These observations suggest an indispensable role of ephrin-B2 in the morphology of the inner ear. In the EphA2 KO mice kidney, pendrin was also observed at the epithelial-epithelial contact sites. Interestingly, Eph/ephrin interaction would presumably occur at the cell-to-cell contact sites. Thus, EphA2 might also be important to exclude pendrin from cell-to-cell contact sites and contributes to the restricted pendrin localization at the apical surface (Fig. 3f). Furthermore, we have also identified two *EPHA2* mutations from patients carrying mono-allelic *SLC26A4* mutation. The induction of mutations, identified in the patients carrying the mono-allelic *SLC26A4* mutation, into EphA2 impaired its binding ability to ephrin-B2 but not ephrin-A1. While the effect of the substitution of pendrin Ala434 to Thr on molecular activity or localization is not known, the substitution of Thr410 to Met is known to result in mis-localization of pendrin from the

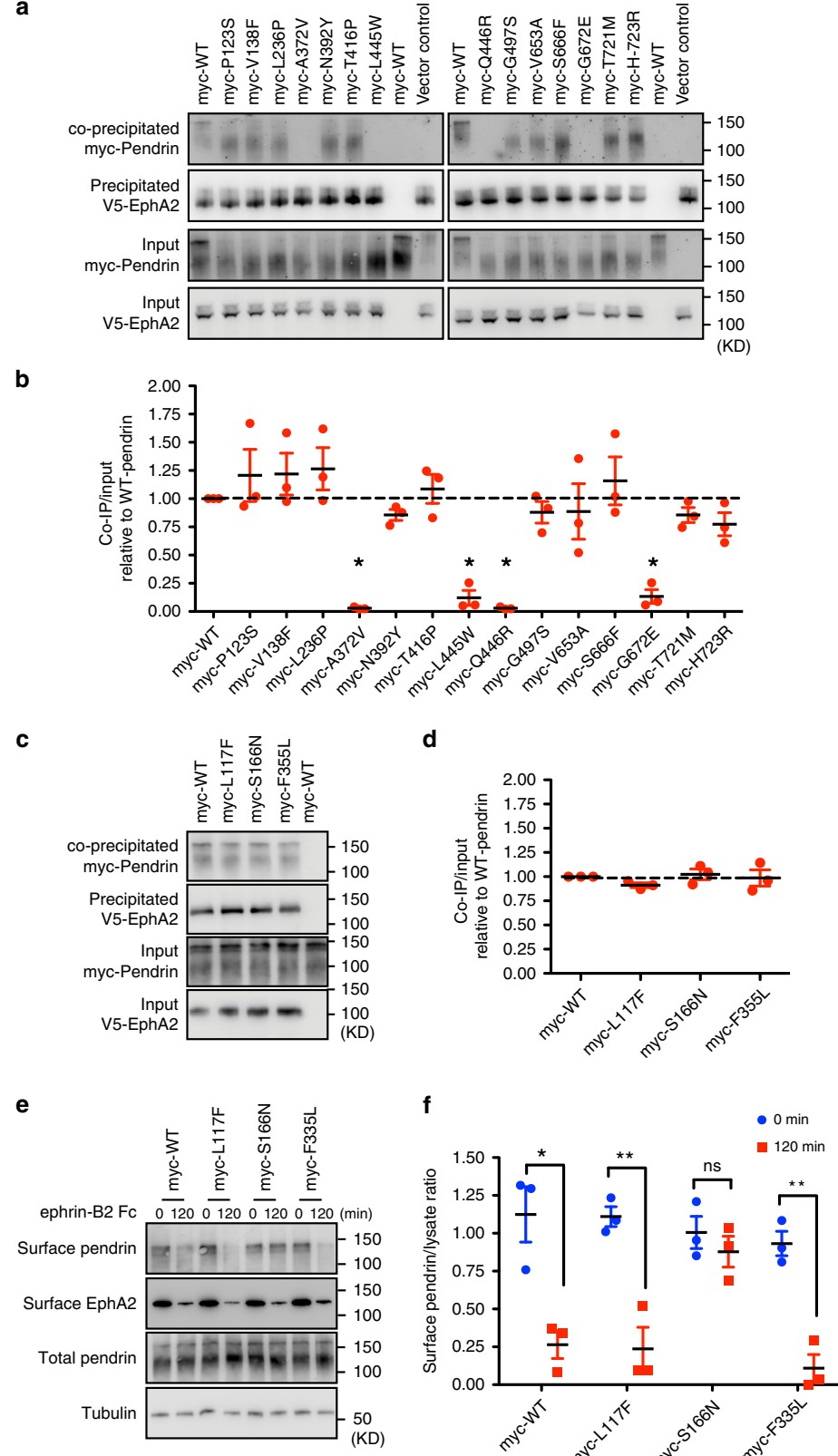

**Fig. 5 Some pathogenic variants of pendrin are not affected by EphA2/ephrin-B2 regulation. a**, **b** Immunoprecipitation of EphA2 with mutated pendrin. myc-pendrin A372V, L445W, Q446R, G672E were not co-immunoprecipitated with EphA2. Densitometric quantifications are shown (**b**). Mean ± SEM; one-way ANOVA with Bonferroni post hoc analyses; *$p < 0.05$; ($n = 3$). **c**, **d** Immunoprecipitation of EphA2 with mutated pendrin. Immunocomplex of myc-pendrin L117F, S166N and F355L was not affected. Densitometric quantifications are shown (**d**). Mean ± SEM; ($n = 3$). **e**, **f** Internalization of EphA2 and mutated pendrin triggered by ephrin-B2 stimulation. Pendrin S166N was not internalized after ephrin-B2 stimulation while EphA2 and other mutated pendrins were not affected. **f** Relative amount of cell surface pendrin is shown. Mean ± SEM; one-way ANOVA; **$p < 0.01$; *$p < 0.05$; ($n = 3$). Source data are provided as a Source Data file.

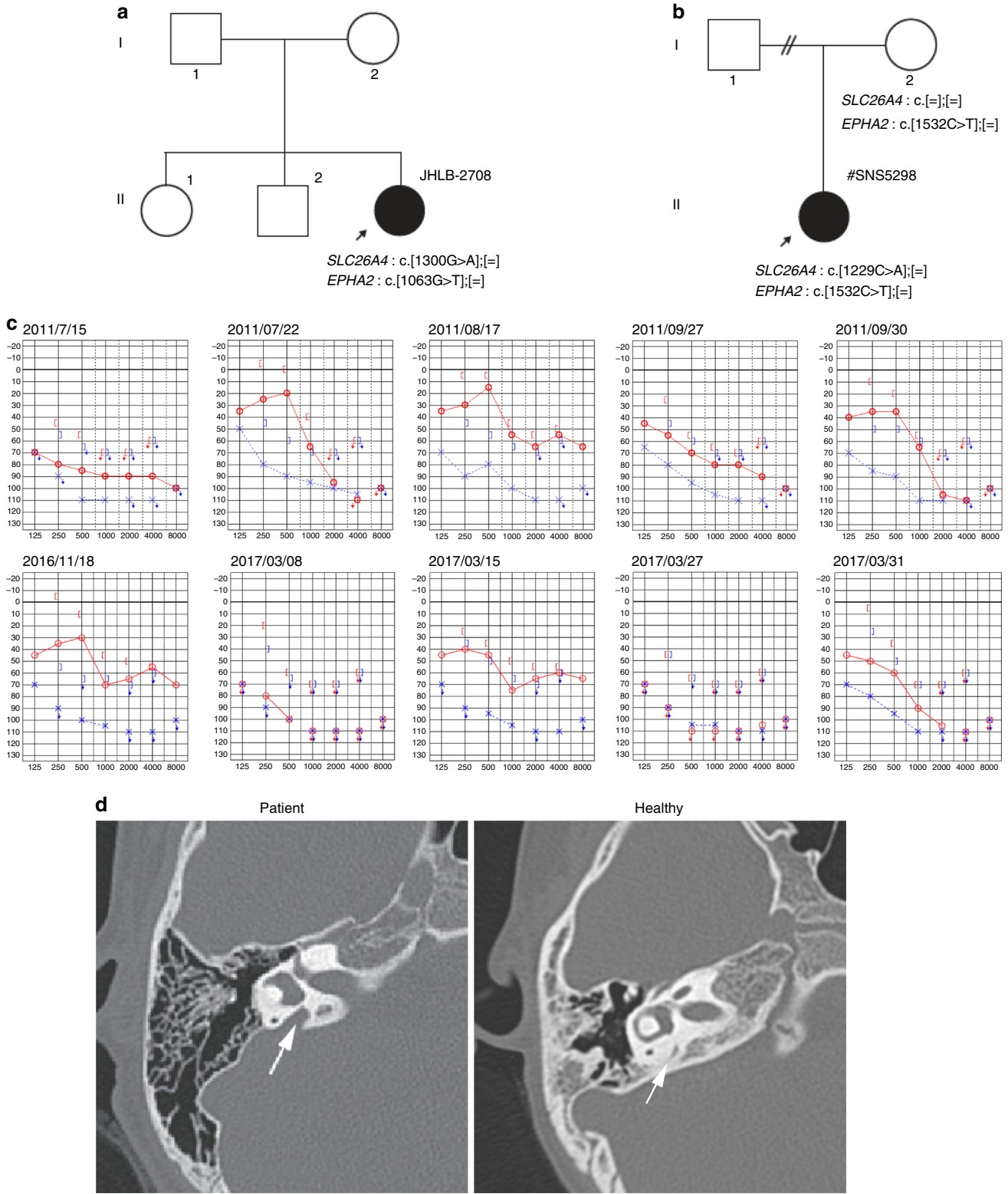

**Fig. 6 Identification and characterization of EphA2 mutation from hearing loss patients with EVA. a, b** Pedigree chart of the patients carrying mono-allelic *EPHA2* and *SLC26A4* mutations. **c** Audiograms of the patient with mono-allelic *EPHA2* p.T511M and *SLC26A4* p.T410M mutations. **d** Temporal bone computed tomography (CT) scan of the patient with mono-allelic *EPHA2* p.T511M and *SLC26A4* p.T410M mutations. The arrow indicates the vestibular aqueduct in the patient and the healthy control.

plasma membrane and loss of iodide efflux ability[6]. We could not access the sequence of the patient's father, but her healthy mother carried the mono-allelic *EPHA2* mutation and did not have any mutation in the *SLC26A4* gene implying a synergistic effect of dual mutations on *EPHA2* and *SLC26A4* on symptoms. Mono-allelic *EPHA2* mutation with the hetero-pendrin mutation may cause pendrin recruitment to the basolateral region in the patients, resulting in the insufficient amount of apical located pendrin and

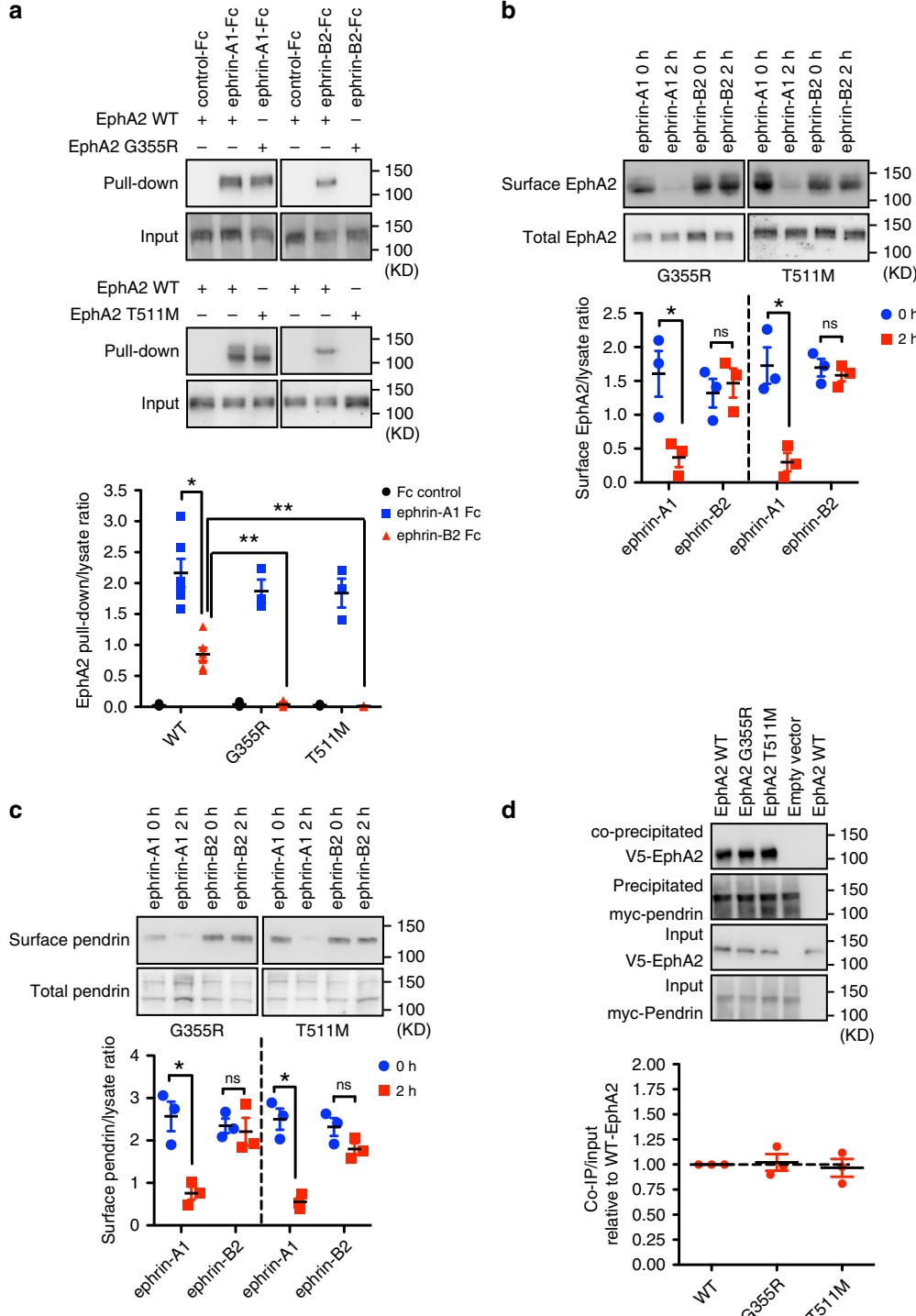

**Fig. 7 Identified EphA2 mutations affect its ability to bind ephrin-B2 but not ephrin-A1. a** Pulldown assay using ephrin-A1 or ephrin-B2 Fc fusion proteins with V5-EphA2 and V5-mutated EphA2. Input and pulldown protein with Fc fusion proteins are shown by immunoblot analysis with anti-V5 antibody. Relative amount of pulled down EphA2 is shown below. Mean ± SEM.; two-way ANOVA; **$p < 0.01$; *$p < 0.05$; (wt, $n = 6$; mt, $n = 3$). **b** The effect of identified mutations on ephrin-A1 and ephrin-B2 induced EphA2 internalization. After surface biotinylation, samples were analyzed by western blotting analysis using an anti-V5 antibody. Relative amount of cell surface EphA2 is shown below. Mean ± SEM; one-way ANOVA; *$p < 0.05$; ($n = 3$). **c** The effect of identified mutations on ephrin-A1 and ephrin-B2 induced pendrin internalization. After surface biotinylation, samples were analyzed by western blotting analysis using an anti-pendrin antibody. Relative amount of cell surface pendrin is shown below. Mean ± SEM; one-way ANOVA; *$p < 0.05$; ($n = 3$). **d** Immunoprecipitation of mutated EphA2 with pendrin. EphA2 mutations do not affect the binding ability of EphA2 with pendrin. Densitometric quantifications are shown below. Mean ± SEM; ($n = 3$). Source data are provided as a Source Data file.

causing functional deficiencies (Supplementary Fig. 8a). Although pendrin localization might be controlled by a tissue-dependent mechanism, our results may suggest at least two potential roles of EphA2 in Pendred syndrome: (1) recruiting pendrin to the cell surface (2) excluding membrane localized pendrin from the basolateral region (Supplementary Fig. 8a). Pendred syndrome is the congenital hearing loss, suggesting the importance of development of the inner ear. The gene deletion of *Efnb2* with a constitutive cre driver line in the inner ear epithelial cells cause malformation at E15.5 likewise *Slc26a4* KO mice[10,20]. Thus, it is likely that EphA2 KO mice may have the inner ear defects at E15.5. Further analysis is warranted.

Given the fact that the identified amino acid substitutions, G355R and T511M are not located in the LBD, it would be interesting to reveal how these two mutations affect ligand-binding specificity between ephrin-B2 and ephrin-A1 to EphA2. As the substitutions replace relatively small residues with bulky residues, it could greatly influence the structural flexibility or rigidity between the surrounding domains. It is worthy noting that EphA2 has been hypothesized to adopt distinct overall structures, "upright" or "flat" with respect to the plasma membrane, depending on cluster density to exert different signaling[49]. Since ephrin-A and ephrin-B are anchored to the plasma membrane in different ways and differ in the distance from the membrane[30], the identified mutations may alter the preference of the overall EphA2 complex structure and may explain the ligand-class-specific effect. Another interesting model to be considered is the hypothetical hexameric form of the EphA4/ephrin-B3 complex reconstructed from crystal lattice assembly, whose distinct asymmetrical conformation at FN1-FN2 linkage necessitates the flexibility in the region (Supplementary Fig. 8b, c)[44], while it would not be necessary for the trimer of the EphA4/ephrin-A5 complex and for the dimer of the EphA2/ephrin-A5 complex (Supplementary Fig. 8d, e). Some protomers of EphA4 need to take on an extended FN1-FN2 conformation while the other protomers bend at FN1-FN2 linkage to alter for all the protomers to attach to the surface of flat membrane (Supplementary Fig. 8c). While oligomerization interfaces are not completely conserved among EphAs and the above model of EphA4 does not necessarily support similar clustering of EphA2-ephrinB2, distinct clustering structures imposed by ephrin-A and ephrin-B and their differential requirement for the flexibility in domain-linkage is another possible scenario.

Some of congenital deafness patients, such as Alport Syndrome, Marshall Syndrome, Stickler Syndrome, and Norrie disease, are associated with cataracts[50–53]. Interestingly, EphA2 is also known as a causative gene of cataracts[54]. It has been shown that homozygous deletion of EphA2 in the KST085 mouse line or the C57Bl/6/129 mouse line backcrossed with the FVB/N background mice are reported to develop progressive cortical cataracts[54]. On the other hand, we did not observe cataracts in our KO mice, which were generated and kept in the C57Bl/6/129 line. This might be due to the different genetic background of mice. Although the two patients carrying *EPHA2* mutation identified this study do not have cataracts, dominant pathogenic variants of human *EPHA2* have been reported and the subjects carrying these variants have cataracts[54]. *EphA2* KO mice are shown to have more susceptibility to microbial infections[55], however, we did not observe any symptom related to this mouse phenotype in the patients. Reported cataract-related *EPHA2* mutations are either located in the intracellular kinase domain or the SMA domain, causing EphA2 loss of function. Conversely identified EphA2 mutations affecting ligand class specificity of ephrins, might cause more restricted effects. Given the broader effect of EphA2 loss of function in such diseases, analysis of *EPHA2* mutations in the patient might be interesting.

## Methods

**Mouse line**. Experiments involving animals were conducted in accordance with institutional guidelines and laws, and following the protocols approved by the local animal ethics committees and authorities (University of Tokyo or Regierung-spraesidium Darmstadt). *EphA2* was targeted in ESCs via promoter trap experiments using the reverse orientation splice acceptor (ROSA) β-Geo retro viral gene trap vector. Insertion site for trap vector is located in the first intron of *EphA2*[23]. Knocked out alleles are shown in Supplementary Fig. 1b. These targeted ESCs were then injected into blastocysts of fertilized C57BL/6 female mice and backcrossed for five generations to generate the *EphA2* knock-out (*EphA2* KO) line. To maintain the *EphA2* knock-out mouse line, C57BL/6 *EphA2* knock-out mice were cross with 129SV mice. Heterozygous transgenic mice did not display any obvious developmental defects. Genotyping PCR amplifications were performed directly on alkaline lysates of tail tips using REDTaq ReadyMix (SIGMA, R2523). The primers used for genotyping were listed in Supplementary Table 2.

**Human subjects and clinical evaluation**. We studied 40 patients affected by congenital sensorineural hearing loss with EVA. Computerized tomography scan was used to diagnose EVA (according to the criteria of EVA: a diameter of >1.5 mm at the midpoint between the common crus and the external aperture), and they were clinically well characterized by repeated auditory examinations. The degree of SNHL ranged from severe or profound. Thyroid function tests were performed for thyroid-stimulating hormone (TSH), free triiodothyronine (FT3), free thyroxine (FT4), and thyroglobulin. The serum concentrations of thyroid hormones were determined using the ECL-IA method. These studies were carried out under the agreement of the patients after ethical approval by the ethics committee of Shinshu University School of Medicine and in accordance with the Declaration of Helsinki. The written informed consent was obtained from those patients.

**Molecular cloning**. pCMV-Myc-pendrin was a kind gift from Professor Min Goo Lee. pcDNA3.2-V5-EphA2 construction was achieved by using Gateway™ LR Clonase™ II Enzyme Mix (Invitrogen, 11791-020) according to the manufacturer's instructions; pDONR223-EphA2 (Addgene, 23926) and pcDNA3.2/V5-DEST (Thermo Fisher, 12489019) were used as donor vector and destination vector. FLAG-HA-pcDNA3.1-EphA2-1-558 and FLAG-HA-pcDNA3.1-EphA2-538-976 were subclone of EphA2 intracellular domain including transmembrane domain (amino acid 1-558) and extracellular domain including transmembrane domain (amino acid 538-976) into FLAG-HA-pcDNA3.1 backbone. The primers used for domain cloning were shown in Suppementary Table 2. Point mutagenesis of pendrin and EphA2 were generated by using KOD Hot Start Mutagenesis kit (Merck Millipore, Toyobo; 71086-3); the pcDNA3.2-V5-EphA2 and pCMV-Myc-pendrin constructs were used as backbones. The primers used for point mutagenesis were listed in Supplementary Table 2.

**Cell culture and transfections**. Madin-Darby Canine Kidney II cells (MDCK II; ATCC #CRL-2936) were cultured in Minimum Essential Medium (MEM; Sigma) with 10% fetal bovine serum (FBS; Biochrom GmbH), 1% Non-Essential Amino Acid (NEAA; Gibco, Invitrogen), 100 IU/ml penicillin and 100 μg/ml streptomycin. Human embryonic kidney 293 (HEK293; ATCC #CRL-1573) cells and human embryonic kidney 293T (HEK293T; ATCC #CRL-3216) cells were cultured in Dulbecco's Modified Eagle Medium (DMEM; Sigma) with 10% FBS, 2 mM L-glutamine (Gibco, Invitrogen), 100 IU/ml penicillin and 100 μg/ml streptomycin.

Plasmids and siRNA transfections into MDCK were carried out with Lipofectamine 2000 (Invitrogen, 11668) using reverse transfection. To knock down EphA2 in MDCK II cells, siRNA AUCACUGCCAAGUUGCCAGAUGCUU was used. Plasmids' transfections into HEK293T were carried out with polyethylenimine (PEI; Polysciences Inc., 24765) transfection protocol. HEK293 T cells were seeded onto 60 mm dish one day before transfection, and cultured with DMEM (10% FBS) for 12 h. Two micrograms of plasimd was added into 200 μl of 150 mM NaCl and immediately mix with 200 μl of 150 mM PEI-NaCl solution (200 μl of 100 mM PEI in 20 ml of 150 mM NaCl). After incubation for 10 min at room temperature, plasmid mixture was then added onto the cells and incubated for 24 h at 37 °C. Cells were analyzed 24 h post transfection.

**Immunoprecipitation from tissues and cultured cells**. Cochleas were dissected from wild type C57BL/6 mice at age of P5. Thyroids and kidneys were dissected from wild type adult C57BL/6 mice. Minced tissues were immersed in 3 ml cold RIPA buffer per gram of tissue (50 mM Tris- HCl, pH 7.4; 1% NP-40; 0.5% sodium deoxycholate; 150 mM NaCl; 1 mM EDTA; Protease inhibitor cocktail, Sigma) and were homogenized on ice. The lysates were then sonicated with ultrasonic homogenizers (BANDELIN) for 5 × 5 s with 50% power. Tissues lysates were centrifuged at 10,000 × g for 15 min and 60 μl of supernatants was taken as "input" and boiled with 30 μl of 3× SDS sample buffer. The supernatants were mixed with 2 μg of anti-EphA2 antibody and 10 μl of dynabeads protein G (EphA2 immunoprecipitation) and incubated with overnight at 4 °C with rotation. Beads were washed with RIPA buffer three times and the immunoprecipitates were analysed with western blotting using specific antibodies.

For cell immunoprecipitation, HEK293T cells were seeded into 60 mm dish, and transfected with pcDNA3.2-V5-EphA2, FLAG-HA-pcDNA3.1-EphA2-1-558, FLAG-HA-pcDNA3.1-EphA2-538-976, pCMV-Myc-pendrin and their corresponding point mutants or control empty vectors using PEI transfection protocol. 24 h after transfection, cells were washed once with ice cold PBS and harvested with lysis buffer (50 mM Tris-HCl, pH 7.4; 1% Triton X-100; 150 mM NaCl; 1 mM EDTA; Protease inhibitor cocktail, Sigma). The lysates were then sonicated with bioruptor (Diagenode) at maximum power for $10 \times 5$ s. Cell lysates were centrifuged at $10,000 \times g$ for 15 min. The supernatants were mixed with 10 µl bed volume of anti-c-Myc magnetic beads (Myc immunoprecipitation), or 15 µl bed volume of anti-FLAG-M2 magnetic beads (Flag immunoprecipitation), and incubated for 1 h at 4 °C with rotation. Beads were washed three times with lysis buffer and the immunoprecipitates were analysed with western blotting.

**Silver staining.** After electrophoresis, gels were placed in 100 ml of fixative solution (50% MeOH, 20% ACOH) for 30 min and washed with 100 ml of 50% ethanol for $3 \times 30$ min. Gels were then incubated with 100 ml of sensitizing solution (0.129 g/l Na$_2$S$_2$O$_3$) for 1 min, and wash for $3 \times 20$ s in 100 ml of ultrapure water. For staining, 100 ml of impregnation solution (AgNO$_3$) was applied to the gels for 20 min. The gels were washed with 100 ml of ultrapure water for $2 \times 20$ s. In all, 100 ml of developing solution (60 g/l Na$_2$CO$_3$; 37% HCOH/I; 4.3 mg/l Na$_2$S$_2$O$_3 \times$ 5 H$_2$O) was added onto the gel and incubate for 5 min at room temperature with gentle agitation on a rotary shaker. Once the desired band intensity was achieved, immediately discard developer solution and add 10 ml of stopper solution (50% MeOH, 10% ACOH) directly to the gel and gently agitated for 10 min. The gel was washed with 100 ml of washing solution B (50% MeOH) for $2 \times 30$ min.

**Surface biotinlaytion.** Ephrin-A1 Fc chimera protein and ephrin-B2 Fc chimera protein were first pre-clustered before adding onto cells. Pre-clustering was achieved using 0.2 µg of goat anti-human IgG Fc (anti-Fc, Jackson Laboratories) per µg of Fc-proteins (at a concentration of 10 µg/ml) for 30 min at room temperature. Clustered Fc proteins were then added onto cells. MDCK II or HEK293T cells were cultured in 60 mm dishes and washed with 3 ml of ice cold washing solution (PBS containing 1 mM MgCl$_2$ and 1 mM CaCl$_2$) for three times. Cells were treated with 1 ml of Sulfo-NHS-biotin (0.5 mg/ml in washing solution, Thermo, #21217) at 4 °C for 1 h. Then cells were applied with 3 ml of stop solution (100 mM glycine in PBS) for three times. To precipitate biotin-labeled surface proteins, cells were treated with 800 µl of lysis buffer (25 mM Tris- HCl, pH 8.0; 1% Triton X-100; 150 mM NaCl; 5 mM EDTA; Protease inhibitor cocktail, Sigma) for 15 min at 4 °C with rotation. Lysates were centrifuged at $10,000 \times g$ for 15 min. 60 µl of supernatants was taken as "in put" and boiled with 30 µl of 3× SDS sample buffer. Aliquots of supernatants were precipitated with 100 µl of streptavidin-coated agarose beads (Invitrogen, S951) at 4 °C for 2 h with rotation. Beads were then washed three times with lysis buffer and proteins were eluted with 60 µl of 1× SDS sample buffer.

**Recombinant Fc proteins pull down.** MDCK II or HEK293T cells transfected with mock, wild type or mutated EphA2 constructs were incubated with pre-clustered ephrin-A1 Fc and ephrin-B2 Fc for 0, 1, and 2 h at 37 °C. The cells were then washed with PBS for 2 times and were harvested with lysis buffer (50 mM Tris-HCl, pH 7.4; 1% Triton X-100; 150 mM NaCl; 1 mM EDTA; Protease inhibitor cocktail, Sigma). The lysates were then sonicated with Bioruptor (Diagenode). The supernatants were mixed with 10 µl of dynabeads protein G (Fc protein precipitation) and incubated for overnight at 4 °C with rotation. Beads were washed with lysis buffer three times and the precipitates were analysed with western blotting using anti-V5 tag antibodies.

**ELISA binding assay.** ELISA binding assay was performed to determine the Kd values[56]. In short, EIA 96 well plates (Costar, Corning Inc.) were coated with ephrin-A5 Fc or ephrin-B2 Fc fusion proteins (3 µg/ml) in 50 mM carbonate buffer pH9.5 overnight at 4 °C, prior to blocking with 5% (w/v) BSA, 0,05% (v/v) Tween-20 in PBS (PBST) for 1 h at room temperature. After three times washes with PBST, diluted EphA2 Fc analyte was added and incubated ro 1 h at room temperature. After three times washes with PBST, plates were incubated with biotinylated 1F7 mAb (1 µg/ml) in PBS for 1 h at room temperature. After three times washes with PBST, UltraStreptavidin-HRP (Thermo Fischer Scientific) dilute 1/500 in PBS was added and incubated 1 h at room temperature. After three times wahs with PBST, final detection via the addition of OPD (SigmaFast$^{TM}$) with the end product was measured at OD492 nm.

**Tissues and cells immunostaining.** For inner ear immunostaining, inner ears were first fixed with 4% paraformaldehyde (PFA) by injection through round window. Next, inner ears were incubated in phosphate buffer (Wako Pure Chemical Industries, 167-14491) containing 10% EDTA overnight at 4 °C, followed by dehydration with 30% sucrose. Samples were then embedded into OCT cryostat sectioning medium and 10 µm crysections were made by Leica cryotome (Leica, CM3050S). Sections were blocked in blocking buffer containing 4% FBS diluted in PBS at room temperature for 1 h. Primary antibodies (anti-EphA2, 1/50, R&D Systems AF1556; anti-pendrin, 1/50, Bio-Rad, 2150-1470; anti-KCNJ10, 1/50, BD

Biosciences, 555289; Phalloidin, Alexa-Fluor-647, 1/500, Invitrogen, A22287; anti-ephrin-B2, 1/100, R&D Systems AF496), diluted in blocking buffer (2% FBS diluted in PBS) were applied overnight at 4 °C. After washing, inner ear sections were incubated with the corresponding Alexa-Fluor-coupled secondary antibody (Invitrogen, 1/500) in blocking buffer (1% FBS in PBS) for 2 h at room temperature. Sections were flat-mounted using Fluromount-G (SouthernBiotech, 0100-01).

For kidney and thyroid immunostaining, mice were perfused with PBS and 4% PFA respectively. The tissues were embedded in OCT and frozen immediately. The rest of staining steps were the same as descried above. Primary antibodies (anti-EphA2, 1/50, R&D Systems AF1556; anti-pendrin, 1/50, Bio-Rad, 2150-1470; lectin Dolichos biflorus agglutinin, 1/500, Invitrogen, A22287) were used for kidney and thyroid immunostaining.

For cells immunostaining, transfected or mock MDCK II cells were first cultured on gelatin coated cover slides. Cells were washed with PBS once and fixed in 4% PFA at room temperature for 10 min. After three times washing with PBS, cells were treated with permeabilization buffer (0.1% Triton-X-100 in PBS) for 15 min and then blocked in blocking buffer (0.1% BSA in PBS) for 30 min at room temperature. Primary antibodies (anti-EphA2, 1/50, R&D Systems AF1556; anti-pendrin, 1/50, Bio-Rad, 2150-1470; anti-EEA1, 1/50, BD Biosciences, 555289; Phalloidin, Alexa-Fluor-647, 1/500, Invitrogen, A22287) were applied overnight at 4 °C. After three times washing, slides were incubated with the corresponding Alexa-Fluor-coupled secondary antibody (Invitrogen, 1/500) for 2 h at room temperature. Slides were flat-mounted using Fluromount-G (SouthernBiotech, 0100-01).

To rescue mutated pendrin localization, 10 µg of V5-EphA2 and 5 µg of myc-pendrin mutants were transfected into MDCK II cell. Non-permealibized cells were stained with c-Myc antibody.

All immunostaining images were taken by Leica TCS SP8 and ZEISS LSM 710 confocal laser scanning microscopes. Images were analyzed by Volocity 6.3 software. The specificity of these antibodies was evaluated as shown in Supplementary Table 3.

**Cell lysis and western blotting.** Cells were rinsed once with PBS, and harvested with 1× SDS sampling buffer (50 mM Tris-HCl pH6.8, 2% SDS, 10% glycerol, 1% β-mercaptoethanol, 12.5 mM EDTA, 0.02% bromophenol blue) and boiled at 95 °C for 10 min. The protein samples were then subjected to immunoblot analysis using indicated antibodies. Uncropped images are shown in Supplementary Fig. 9.

**Mass spectrometry.** Immunoaffinity purifications by anti-EphA2 antibody and control IgG from adult wild type C57BL/6 mice kidneys were subjected in parallel to In-gel digestion[57]. Five gel blocks were cut per sample and peptides finally captured, cleaned and stored using STAGE tips[58].

For mass spectrometry (MS) analysis, peptides were eluted from STAGE tips by solvent B (80% acetonitrile, 0.1% formic acid), dried down in a SpeedVac Concentrator (Thermo Fisher Scientific) and dissolved in solvent A (0.1% formic acid). Peptides were separated using a nano-flow HPLC system (EASY-nLC, ThermoFisher Scientific) and 20 cm, in-house packed C18 silica columns (1.9 µm C18 beads, Dr. Maisch GmbH) coupled in line to a QExactive orbitrap mass spectrometer (ThermoFisher Scientific) using an electrospray ionization source. Peptides were subjected to a linearly increasing gradient concentration of solvent B (90% acetonitrile, 1% formic acid) over solvent A (5% acetonitrile, 1% formic acid) from 10% to 38% for 55 min and from 38% to 60% for 5 min, followed by washing with 95% of solvent B for 5 min and re-equilibration with 5% of solvent B. Full MS spectra were acquired in positive mode and a mass range of 300 to 1650 m/z with a resolution of 70,000 at 200 m/z. The ion injection target was set to $3 \times 10^6$ and the maximum injection time limited to 20 ms. Ions were fragmented by high-energy collision dissociation (HCD) using an isolation window width of 1.8 m/z, a normalized collision energy of 27 and a ion injection target of $5 \times 10^5$ with a maximum injection time of 120 ms. The resulting tandem mass spectra (MS/MS) were acquired with a resolution of 35,000 at 200 m/z using data dependent mode with a loop count of 10 (top10). The instrument was further configured to use a minimum AGC target of $5 \times 10^2$ and an intensity threshold of $4.23 \times 10^3$ for fragmentation spectra and exclude precursor ions with a charge state of 1, as well as unassigned charge states and members of already targeted isotope clusters and to use a dynamic exclusion time of 20 s.

Mass spectrometric raw data were caluculated using the MaxQuant/Andromeda suit of algorithms (v. 1.5.2.8)[59]; against the human UniprotKB database (88703 entries; The UniProt Consortium, 2007), complemented with the sequence of murine EphA2 using the default parameters, including a minimum required length of seven amino acids per peptide identification, a minimum of one peptide per protein group identification, carbamidomethylated cystein as a fixed, as well as oxidated methionine and n-terminal protein acetylation as dynamic modifications. The match-between-runs feature for ID transfer was enabled. Using the target/decoy database approach[60], the false discovery rate was controlled at 1% for the peptide and protein groups levels. MaxQuant's label free quantitation algorithm[61] was used to compare bait and control experiments.

**Hemotaxylin and Eosin staining.** Hemotaxylin and Eosin (H&E) staining was used for tissues morphological analysis. Mice kidneys, thyroids and inner ears were

harvested and dehydrated by ethanol, xylene and paraffin. Tissues were embedded for paraffin sectioning with 10μm tissue sections cut by microtome (Thermo, HM355S). To deparaffinize and rehydrate, sections were first immersed in xylene for 3 × 10 min, followed by serial ethanol treatment with 100%, 95%, and 80% ethanol for 3 × 5 min. Slides were then immersed in deionized water for 5 min. For hematoxylin staining, tissues were incubated with hematoxylin for 3 min and rinse in deionized water. Slides were then immersed in tap water to allow staining to develop. For eosin staining, slides were incubated with eosin for 30–45 s and then followed by 95 and 100% ethanol washing for 3 × 3 min. After final washing with xylene for 3 × 5 min, slides were mounted with mounting medium. Nanozoomer (Hamamatsu Photonics) was used for visualization of H&E staining slides. Thickness of inner ear stria vascularis and area of thyroid follicular lumen were measured by NDP.viewer.2 software.

**X-ray microtomography**. Mice inner ears were isolated and pretreated with 4% PFA. After washing with PBS, inner ear samples were scanned using SkyScan 1276 microCT imaging system (Bruker). In all, 6424 projection images were acquired over a 0.5 mm filter with the X-ray source set to 50 kV over a complete 360° rotation for 1 h. Images were reconstructed using NRecon software. Image processing software CTVox was used to generate 3D images and CTAn was used to measure inner ear cochlear area.

**Genetic diagnosis**. Genetic screening was performed in two cases using an Invader assay to screen for 68 known hearing loss-related mutations and direct sequencing for *EPHA2* and *SLC26A4* mutations.

**Protein structures analysis**. Protein domains were analyzed using Pfam[62]. 3D protein structures were predicted on the SWISS-MODEL and Phyre2 servers[63].

**Statistical analysis**. All statistical analyses were carried out using Prism software (Graph Pad, CA). A $P < 0.05$ was considered statistically significant. Data are based on at least three independent experiments or three mutant and control animals for each stage.

**Reporting summary**. Further information on research design is available in the Nature Research Reporting Summary linked to this article.

## Data availability

The mass spectrometry proteomics data have been deposited in the ProteomeXchange Consortium via the PRIDE partner repository under the accession code PXD017011. The source data underlying Figs. 2b, e, g, 3e, 4e, g, h, 5b, d, f, 7a–d and Supplementary Figs. 2b, 3a, b, d, f, 5b, and 7d are provided as a Source Data file. The remaining data are available within the article and its supplementary information files and from the corresponding author on reasonable request.

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

## Acknowledgements

Funding for this project was provided by the Excellence Cluster Cardio-Pulmonary System, the German Research Foundation, Deutsche Forschungsgemeinschaft (GRK2213), and Fritz Thyssen Stiftung (Az. 10.14.2.178). M.G.L. was supported by funding from NRF-2013R1A3A2042197 from the National Research Foundation, the Ministry of Science and ICT, Republic of Korea.

## Author contributions

M.L., S.N., S.U., and M.N. designed the study. C.N., M.G.L., and M.A. provided materials including EphA2 KO mice. M.L., T.K., T.B., and S.K., examined the experiments using the inner ear. M.L., S.S., A.W., and S.O. examined μCT analysis. F.S., L.C., B.W.D., and A.B. examined KD values of the ephrin-B2/EphA2 complex. M.K. and J.G. analysed mass spectrometry-based proteomics data. Pendred syndrome patients were analysed by S.N., S.K., and S.U. All other experiments were performed by M.L., M.R., F.M., and T.H. M.L. and M.N. wrote the paper.

## Competing interests

The authors declare no competing interests.
