## [Peer Review File · Nature Communications]

Reviewers' comments:

Reviewer #1 (Remarks to the Author):

The authors of this study investigate the role of the receptor tyrosine kinase EphA2 in the aetiology of Pendred syndrome with a link to pendrin and its associated hearing loss. The authors have previously shown an involvement of the EphB/ephrinB family in vascular morphogenesis which controls internalization of the VEGF receptor. Here ephrinB2 loss results in abnormal development of the inner ear.

Here the authors demonstrate that ephA2 knockout mice have a dis-organized development of the inner ear, and the authors present a set of data indicating that a ligand-induced internalization of EphA2 with pendrin is important for pendrin function and proper hearing, that is mutations interfering specifically with an ephrinB2/EphA2 but not an ephrinA1/EphA2 interaction are found in patients with pendred syndrome.

While the model overall presented is highly interesting, there are massive technical issues with the data which have to be substantially rectified before a publication can be even considered.

The authors indicate an interaction of pendrin with EphA2, using co-immunoprecipitation. The western blots shown in Fig. 1 here are not of good quality and do not allow this conclusion. Also, suitable controls are missing.

I am not convinced by data shown in Fig 4, indicating that ephrinB2 is an (atypical) ligand for EphA2. In particular the lack of tyrosine phosphorylation of EphA2 after treatment with ephrinB2 is a strong indication that this interaction is physiologically not relevant. Overall, the positive control with ephrinA1 indicates that the experiments have been carried out well.

The differential internalization of pendrin in response to ephrinA1 or ephrinB2 is difficult to interpret since a quantification and statistical analysis is not presented. Indeed, in general western blot data presented in the manuscript are not quantified and not statistically characterized.

Reviewer #2 (Remarks to the Author):

The manuscript from M. Li et al., entitled "EphA2 is an additional causative gene of Pendred syndrome with pendrin" that was submitted to Nature Communications by Dr. Nakayama reports several interesting findings about a possible second gene associated with enlarged vestibular aqueduct (EVA), a common radiological finding in children often associated with sensorineural loss of hearing. About 25% of cases of EVA is due to biallelic pathogenic mutations (M2) of SLC26A4. However, 25% of EVA subjects have only one pathogenic

variant (M1) in the coding exons of SLC26A4, while 50% have no pathogenic variants (M0) of SLC26A4 coding sequence. In the manuscript from Li and co-authors, data is provided that EPHA2 protein interacts/partners with pendrin and heterozygous variants of human EPHA2 and heterozygous variants (M1) of SLC26A4 can give rise to EVA (an example of digenic inheritance). These observations provide an explanation for two M1 cases in Japan. The authors also characterized an Epha2 knockout mouse (see comments below).

Overall, the study from Li and co-authors has potential to provide valuable new insight about a complex of EPHA2 and pendrin, further mechanistic insight about pendrin function and another cause of EVA in some patients. However, there are significant experimental flaws and omissions in the text and figures. The Introduction and Discussion are missing published studies that should be known to the authors, are directly relevant to the topics in this manuscript. The findings in these papers, omitted from this manuscript for unknown reasons, may require revision of the interpretation of some of the authors' data. There are also minor issues that need attention.

Major issues:

1. Figure 1 panel "e" utilizes an Epha2 knockout mouse. How was this mutant mouse engineered? It is incompletely described in the Methods section on page 16. Jun et al., (2009, PLoS Genetics volume 5, e1000584) reported that a "homozygous deletion of Epha2 in two independent strains of mice developed progressive cortical cataract." Khounlotham et al., 2009 reported that homozygous Epha2 mutant mice are more susceptible to microbial infections. Does the Epha2 mutant mouse described in the Li et al., manuscript have cataracts? Why were these published Epha2 mouse models and their phenotypes not mentioned in the manuscript under review?
2. The authors indicate that in their mutant mouse model, "Epha2 was targeted in ESCs via homologous recombination to replace the entire coding region of Epha2 with a lacZ cassette, resulting in a targeted deletion of EphA2." This statement is confusing. Which exons of EPHA2 were deleted? The forward primer used to genotype the "EphA2 mt" allele provided on page 16 is located in intron 1 (i.e. between exon 1 and exon 2 of EPHA2). In part, exon 1 of EphA2 is protein coding. If the wild type EphA2 exon 1 was retained, the entire coding region of EPHA2 was not deleted in the Li et al., ko mouse. Please provide a schematic drawing illustrating exactly how the EphA2 ko was engineered and the locations of the PCR primers that were used to genotype their mouse and the wild type allele.
3. For the data in panel d of Figure 1, add the kDa's for each of the proteins in the various Western blot slices that are shown in this Figure and in Figures 4 and 5. Additionally, provide the full Western blots in as supplementary data. Similarly, provide images of the full blots for all of the other Western blots in this study where only a slice of the Western is shown.

Additionally, add a section in the results or methods section describing the data validating the specificity of all the antibodies used in this study. Indicate in all legends the number of independent experiments and technical replicas that were conducted for each Western.

4. Figure 6, panels a and b show two hearing impaired EVA individuals who are double heterozygotes for a variant of EPHA2 and a variant of SLC26A4. The authors may be correct that this combination of two non-allelic heterozygous recessive variants, one in EPHA2 and one in SLC26A4, cause EVA. If correct this would be an important finding. So, let's examine the evidence. The EPHA2 of 41 individuals with EVA (and having one coding region pathogenic variant of SLC26A4) were sequenced and 2 of 41 were found to be carrying heterozygous variants of EPHA2. Several in silico predictions indicating these variants of EPHA2 are damaging (the specific algorithms were not mentioned (should be) and scores were not provided, for example, CAAD scores). Were any of the in silico predicted not to be damaging?

For the two affected individuals in figure 6, Whole Exome Sequencing (WES) should be conducted to rule out biallelic mutations of one of the already reported ~70 to 80 human nonsyndromic deafness genes as the cause of hearing loss in these patients. Understandably, the two individuals have EVA and are monoallelic for a SLC26A4 variant. Li and co-authors may be correct that they have documented an example of digenic inheritance of EVA. But more rigor would be important in ruling out a simpler explanation for the hearing loss and EVA since you don't have linkage data. Consider doing whole exome sequencing (WES) on the two deaf individuals. WES analyses have become affordable and wouldn't take long to accomplish.

5. In the OMIM database, dominant pathogenic variants of human EPHA2 have been reported and the human subjects carrying these variants have cataracts. The title of the Jun et al., 2009 paper is "EPHA2 is associated with age-related cortical cataract in mice and humans." There is no mention of these reported observations in the manuscript from Li et al. Why is that? Do the two individuals (provide their ages) with monoallelic variants of EPHA2 in Figure 6 have cataracts?

6. Chattaraja et al., 2017 reported that in M1 subjects (one coding mutation of SLC26A4), the second allele in trans is a haplotype of 12 uncommon variants upstream of SLC26A in two independent cohorts, one from the United States and a second from Denmark. The EVA-associated haplotype (called CEVA) is common but apparently not in Asian populations. This study was not mentioned in Li et al.,. Do the two of 40 EVA subjects with one variant of EPHA2 and one coding variant of SLC26A4 have the CEVA haplotype? In the Chattaraja et al., 2017, CEVA appears to be the missing variant in most of their M1 EVA subjects. CEVA was reported to be rare in Asian populations but may not be entirely absent. 2 of 41?

7. On page 16, there is no statement that all of the 41 human subjects in the study by Li and co-authors were ascertained after approval by a Helsinki committee or IRB and that the subjects signed an informed consent for participation in the study and the use of their

genomic DNA. Was the study of human subjects in Li et al. formally approved by an institution where the authors are located? Please add this information. If not, discuss this issue with the editors of Nature Communications who will consider how this might impact publication.

Minor issues are described below first for the, title, figures and then in order of appearance in the text.

The title of the manuscript “EphA2 is an additional causative gene of Pendred syndrome with pendrin” is strange. “... Pendred syndrome with pendrin” is ambiguous. In the title, there is a gene name (Epha2), then the name of a syndrome (Pendred) followed by the name of a protein (pendrin). Think up a better title for your manuscript. You may well have an example of digenic inheritance of EVA.

Figure 1, panel b, what is the column labeled “Pass”? Nothing mentioned in the legend about “Pass”. Add size standards to images in panels a and b. What is your interpretation of all the bands in panel b, second column?

Figure 2, panel b, the stria is thinner in the Epha2 ko. Can you tell which of the three cell layers (marginal, intermediate, basal) of the stria contribute to the thinning quantified nicely in panel c? In Figure 3, KCNJ10 staining is reduced. In what cell layer(s) of the stria vascularis is KCNJ10 known to be located? Since the stria is thinner compared to the wild type, is the endocochlear potential (EP) also reduced?

Figure 5, the variants of SLC26A4 were reported by others and should be referenced. In the legend, add a “the” between “from” and Plasma membrane” Also, “Some disease forms of pendrin...” Do you mean by “forms”, mutant alleles/pathogenic variants?

Figure 6, the arrow in the CAT scan points to an enlarged vestibular aqueduct. Mention this in the legend.

Results, page 6, add “an” between “.... carried out..” and “...interactome analysis.”

Page 13, delete “to” after “... membrane is not...”

Reviewer #3 (Remarks to the Author):

The manuscript by Li and coworkers reports on the role of EphA2 in hearing loss patients including Pendred syndrome. The results suggest that EphA2 binding to pendrin, a transporter protein that is causally linked to Pendred syndrome, may be essential for proper plasma membrane recruitment of pendrin.

While this reviewer acknowledges the potentially interesting and novel aspects of such a mechanism, the presented data seem rather preliminary and do suffice to support such a conclusion. Below, I am listing specific points of criticism in chronological order.

1. The evidence for direct binding of pendrin to EphA2 in vivo is not very strong. Co-IP expts as in Fig. 1d are not well controlled. Since the authors have access to EphA2^{-/-} mice, similar Co-IP expts should be done with EphA2^{-/-} tissue lysates to exclude the possibility that pendrin is brought down with anti-EphA2 Abs unspecifically.

2. EphA2^{-/-} mice have inner ear abnormalities. I find the morphological analysis unsatisfying. In Fig. 2a, the size of the cochlear duct seems “slightly enlarged”. Such a statement needs to be supported with quantified data. In Fig. 2b, the thickness of the stria vascularis is reduced. It is unclear from the images how and where the thickness was measured. As a minor point, in Fig. 2c, the y axis should always start at 0 μm . In Fig. 2e, the number of mice for the quantification was not indicated.

3. In Fig. 3, pendrin mislocalization in EphA2^{-/-} mice is fairly convincing. However, in Fig. 3d, the reduction of KCNJ10 immunoreactivity in EphA2^{-/-} mice should be quantified in groups of n=3 mice.

4. The authors claim that ephrinB-B2 is an atypical ligand of EphA2. Since this is against current models of ephrin/Eph interactions, the authors need to provide better support for this claim. Biophysical interaction studies, e.g. using Biocore, are needed in comparison with EphA4. In Fig. 4b,c, the Co-IP expts are not well controlled. Anti-ephrin-B2 IPs may bring down many proteins unspecifically. In Fig. 5e, the rate and timing of EphA2 internalization should be quantified. Similar expts in Fig. 5d look very different. While the rate of EphA2 internalization seems rather different between ephrinB-B2 and A1, the rate of pendrin internalization looks exactly the same (although again not quantified). The conclusion that both events are causally linked requires more experimental support.

5. In Fig. 5, the regulation of pendrin localization by EphA2 is preliminary. It is unclear to me why disease forms of pendrin were co-IPed with a truncated version of EphA2 consisting only of the cytoplasmic and transmembrane domains. First of all, interactions of pendrin mutants with full-length EphA2 should be investigated, and the data should be quantified across several blots. Second, cytosolic mutants of pendrin that bind to EphA2 should be further analyzed, e.g. using immunofluorescence imaging. Co-expression with EphA2 should recruit (at least some of) those mutants to the plasma membrane, and ephrin stimulation should exclude them again from the PM. In Fig. 5c, it is unclear what is depicted in left and right panels. In Fig. 5d, the rates of pendrin and EphA2 internalization should be quantified.

6. In Fig. 6d, the patient CT scan is not useful unless a similar scan from a healthy person is shown.

7. In Fig. 7, the analysis of EphA2 mutations is incomplete and confusing. The biggest

problem is that these EphA2 mutations do not affect EphA2's ability to bind wild-type pendrin (Fig. 7e). Hence, the mechanism by which these mutations affect inner ear function remains unclear. Perhaps, the association between EphA2 and pendrin is weaker? The authors did not perform expts as in Fig. 4e (pendrin internalization after ephrinB stimulation). Moreover, the blots shown in Fig. 7c,d should be quantified.

We are very grateful to all of the reviewers for their positive evaluation of the paper, and would like to thank them for their constructive feedback working towards improving our manuscript. As you will see, we have extensively revised the manuscript by adding a large amount of new data. While a detailed point-by-point response to all comments is given further below, here is a brief summary of the most important additions.

1. All of the western blots suggested to be quantified in the comments were quantified and statistically analyzed.
2. The western blotting showing immunocomplex of EphA2, ephrin-B2 and pendrin was repeated and replaced with the improved images. In addition, the tissues from EphA2 KO mice were employed as a negative control.
3. For better data presentation, the inner ear structures of mutant mice were shown by μ CT with quantification.
4. To further investigate the role of EphA2 as a receptor for ephrin-B2, we measured the Kd value of ephrin-B2/EphA2 by ELISA binding assay. Additionally, we re-examined the auto-phosphorylation of EphA2 triggered by ephrin-B2 in a longer time course by western blotting. ephrin-B2 induces EphA2 auto-phosphorylation while its effect is weaker than that of ephrin-A1. Furthermore, we showed that EphA2 was colocalized with an early endosome marker, EEA1 after ephrin-B2 stimulation.
5. To avoid confusion, we mentioned detailed patient characteristics used for sequence analysis and cited a previously reported manuscript. We exclude the possibility that two Japanese patients carry mutations on the reported genes as a risk factor of hearing loss. Also we exclude the possibility that those two patients have the CEVA haplotype.

Thus, the sum of all the data in the manuscript now strongly support that EphA2 plays important roles in the inner ear function together with pendrin, and thereby being a critical factor of pendred syndrome.

Furthermore, we established ephrin-B2 as a functional ligand of EphA2. Interestingly, identified EphA2 mutations in the two Japanese patients compromised binding ability of EphA2 with ephrin-B2 but not ephrin-A. As Eph receptors and ephrins are involved in the regulation of many other biological and pathological processes, our findings may well have much wider significance.

Please find a point-by-point response below:

Reviewer #1:

Reviewer: The authors indicate an interaction of pendrin with EphA2, using co-immunoprecipitation. The western blots shown in Fig. 1 here are not of good quality and do not allow this conclusion. Also, suitable controls are missing.

Reply: We appreciate this comment to improve the quality of our manuscript. We re-examined immunoprecipitation with an EphA2 antibody from the tissue lysates. Importantly, the tissues from EphA2 KO mice were used as a negative control (Fig. 1d). Pendrin was co-immunoprecipitated with EphA2 from tissue lysate from control mice but not that from EphA2 KO mice. Full scan of the images was included in the Sup Fig. S9.

Reviewer: I am not convinced by data shown in Fig 4, indicating that ephrinB2 is an (atypical) ligand for EphA2. In particular the lack of tyrosine phosphorylation of EphA2 after treatment with ephrinB2 is a strong indication that this interaction is physiologically not relevant. Overall, the positive control with ephrinA1 indicates that the experiments have been carried out well.

Reply: Thank you for pointing out the very important issue. It was believed that ephrin-B2 is not the ligand of EphA2. Thus, we carefully re-investigated the relationship between ephrin-B2 and EphA2.

First of all, ephrin-B2 was co-immunoprecipitated with EphA2 by using an anti-EphA2 antibody from tissue lysate of control mice, but not of EphA2 KO mice (Fig. 4b), further supporting our conclusion.

Secondly, we calculated the K_d value between the ephrin-B2 and the EphA2. The fc-fusion protein of the EphA2 extracellular domain was immobilized onto the ELISA plate and binding of that of ephrin-B2 was examined. The K_d value between ephrin-B2 and EphA2 was 85nM, while that between EphA2 and ephrin-A5, the classic ligand of EphA2 was 2nM. The binding of ephrin-B2 with EphA4 as a ligand has been analyzed. Especially, a previous observation by Bowden et al. (2009, Structure volume 17, 1386-1397), has determined the K_d values of EphA4/ephrin-B2, EphA4/ephrin-A4 and EphA4/ephrin-A5 respectively. (EphA4/ephrin-B2, 10.8μM; EphA4/ephrin-A4, 360nM; EphA4/ephrin-A5, 36nM). The K_d value of EphA4/ephrin-B2 was approximately 30 times greater than that of EphA4/ephrin-A5 and 300 times greater than that of EphA4/ephrin-A4. Thus, the K_d value of EphA2/ephrin-B2 was within a reasonable range.

In the original manuscript, we have shown that ephrin-B2 Fc induces EphA2 internalization one hour after ephrin-B2 stimulation. On the other hand, EphA2 auto-phosphorylation was examined up to 30 minutes after stimulation. Thus, we examined the effect of ephrin-B2 stimulation of EphA2 auto-phosphorylation in a longer time course. Stimulation of EphA2 with ephrin-B2 Fc induced auto-phosphorylation of EphA2 in 15 minutes, which was sustained at least up to 2 hours. The level of EphA2 auto-phosphorylation induced by ephrin-B2 was weaker than that by ephrin-A1, however, treatment of human Fc with the cells failed to induce EphA2 autophosphorylation (Fig. 4d, e).

After stimulation with ligands, receptor tyrosine kinases (RTKs) are internalized and transported to early endosome. Consistently, we observed co-localized immunoreactive signal

corresponding to ephrin-B2, phospho-EphA2 and EEA1, a marker of early endosome, 1 to 2 hours after ephrin-B2 stimulation (Fig. S3e, f).

Taken together, our results indicate that ephrin-B2 is a functional ligand of EphA2 to control membrane localization.

Reviewer: The differential internalization of pendrin in response to ephrinA1 or ephrinB2 is difficult to interpret since a quantification and statistical analysis is not presented. Indeed, in general western blot data presented in the manuscript are not quantified and not statistically characterized.

Reply: We are sorry for all of the difficulties in interpreting data caused by lacking of quantifications. Accordingly, we included quantification and statistical analysis of western blotting analysis (Fig. 4d-h, Fig. 5, Fig. 7, Fig. S3c, d, Fig. S7).

Reviewer #2:

Reviewer: Figure 1 panel “e” utilizes an EphA2 knockout mouse. How was this mutant mouse engineered? It is incompletely described in the Methods section on page 16. Jun et al., (2009, PLoS Genetics volume 5, e1000584) reported that a “homozygous deletion of EphA2 in two independent strains of mice developed progressive cortical cataract.” Khounlotham et al., 2009 reported that homozygous EphA2 mutant mice are more susceptibility to microbial infections. Does the EphA2 mutant mouse described in the Li et al., manuscript have cataracts? Why were these published EphA2 mouse models and their phenotypes not mentioned in the manuscript under review?

Reply: We highly appreciate your comments to improve the quality of our manuscript. We are sorry for not providing detailed information about generation of the EphA2 mutant mouse. Our mice were generated by Naruse-nakajima et al (Mech Dev 2001). To avoid any confusion, the schematic figure showing the targeting strategy is included in Fig. S1b.

Regarding development of cataract and microbial infection, we did not observe those phenotypes in our EphA2 loss of function mice. As some of congenital deafness patients associate with cataract, we carefully observed KO mice lens. However, we did not observe any obscure in mice’s lens or any mis-oriented fiber cells. This may be due to the different gene deletion strategy or genetic background of mice. As the reviewer pointed out, homozygous deletion of EphA2 in the KST085 mouse line and the C57Bl/6/129 mouse line backcrossed with FVB/N background developed progressive cortical cataract, and homozygous EphA2 mutant 129SV mice line are more susceptibility to microbial infections. On the other hand, our EphA2 knock out mouse were generated in C57Bl/6/129 line. Thus, the different genetic background can be the cause of the differences.

As to the susceptibility to microbial infections, unfortunately we did not have chance to examine it. Maintenance of experimental animals in the Max Planck institute are in microbe-free condition under strict monitoring. To perform such experiments related to infections we need to obtain animal experiment permission according to the local law, which is unlikely to be approved in our animal facility. We noticed that infections and inflammations could also affect inner ear and cause deafness (Nakanishi et al. 2017, PNAS volume 37 E7766-E7775. and Nance et al. 2006, Journal of Clinical Virology volume 35 221-225). However, our *in vitro* and *in vivo* experiments together demonstrated EphA2 regulates pendrin surface expression independent with microbial infections. Hence, we believe pendrin mis-localization

is the main reason why the mice inner ears were affected. Of note, the patients carrying EphA2 mutation do not suffer from cataract. Given the importance of these information, we discussed these points in the revised manuscript (p18 line 19 - p19 line 10).

Reviewer: The authors indicate that in their mutant mouse model, “Epha2 was targeted in ESCs via homologous recombination to replace the entire coding region of Epha2 with a lacZ cassette, resulting in a targeted deletion of EphA2.” This statement is confusing. Which exons of EphA2 were deleted? The forward primer used to genotype the “EphA2 mt” allele provided on page 16 is located in intron 1 (i.e between exon 1 and exon 2 of EPHA2). In part, exon 1 of EphA2 is protein coding. If the wild type EphA2 exon 1 was retained, the entire coding region of EphA2 was not deleted in the Li et al., ko mouse. Please provide a schematic drawing illustrating exactly how the EphA2 ko was engineered and the locations of the PCR primers that were used to genotype their mouse and the wild type allele.

Reply: As described in previous comments, we included the schematic figure showing the targeting strategy with primer information (Fig. S1b, table2).

Reviewer: For the data in panel d of Figure 1, add the kDa’s for each of the proteins in the various Western blot slices that are shown in this Figure and in Figures 4 and 5. Additionally, provide the full Western blots in as supplementary data. Similarly, provide images of the full blots for all of the other Western blots in this study where only a slice of the Western is shown. Additionally, add a section in the results or methods section describing the data validating the specificity of all the antibodies used in this study. Indicate in all legends the number of independent experiments and technical replicas that were conducted for each Western.

Reply: We appreciate your suggestions to improve the quality of our data. Accordingly, these points were fixed in our revised manuscript (Fig. 1, 4, 5,7, Fig. S1, S3, S7). Full-blots are shown in Fig. S9.

Reviewer: Figure 6, panels a and b show two hearing impaired EVA individuals who are double heterozygotes for a variant of EPHA2 and a variant of SLC26A4. The authors may be correct that this combination of two non-allelic heterozygous recessive variants, one in EPHA2 and one in SLC26A4, cause EVA. If correct this would be an important finding. So, let’s examine the evidence. The EPHA2 of 41 individuals with EVA (and having one coding region pathogenic variant of SLC26A4) were sequenced and 2 of 41 were found to be carrying heterozygous variants of EPHA2. Several in silico predictions indicating these variants of EPHA2 are damaging (the specific algorithms were not mentioned (should be) and scores were not provided, for example, CAAD scores). Were any of the in silico predicted not to be damaging?

Reply: Thank you for pointing out very important points. In the revised manuscript, in silico predictions are included (Table 1).

Reviewer: For the two affected individuals in figure 6, Whole Exome Sequencing (WES) should be conducted to rule out biallelic mutations of one of the already reported ~70 to 80 human nonsyndromic deafness genes as the cause of hearing loss in these patients. Understandably, the two individuals have EVA and are monoallelic for a SLC26A4 variant. Li and co-authors may be correct that thwy have documented an example of digenic inheritance of EVA. But more rigor would be important in ruling out a simpler explanation for the hearing loss and EVA since you don’t have linkage data. Consider doing whole exome

sequencing (WES) on the two deaf individuals. WES analyses have become affordable and wouldn't take long to accomplish.

Reply: As the reviewer pointed out, reported ~70 to 80 human nonsyndromic deafness genes are identified as the cause of hearing loss. The mutations of those genes in the patients analyzed in our manuscript were previously examined. We could not identify any of the mutations in those genes (Kitano et al. 2017, Plos One volume 12 e0177636). This is a very important point. Thus, we added this information in our revised manuscript (p13 line 8 to10).

Reviewer: In the OMIM database, dominant pathogenic variants of human EPHA2 have been reported and the human subjects carrying these variants have cataracts. The title of the Jun et al., 2009 paper is "EPHA2 is associated with age-related cortical cataract in mice and humans." There is no mention of these reported observations in the manuscript from Li et al. Why is that? Do the two individuals (provide their ages) with monoallelic variants of EPHA2 in Figure 6 have cataracts?

Reply: As described above, the patients carrying EPHA2 mutation, 22 and 14 years old respectively, do not have cataracts. Interestingly, all reported cataracts-related EphA2 mutations are either located in intracellular kinase domain or SAM domain and causing subcellular retention or losing protein stability. On the other hand, the two EphA2 mutations identified in this manuscript are located at the link region of extracellular domains, affecting ligand binding specificity of EphA2. Moreover, we have shown that ephrin-B2 is a ligand of EphA2 in addition to ephrin-As, the classic ligands of EphA2. Given the fact that ephrin-B2 inner ear epithelial cell specific KO mice exhibit an abnormal structure of the inner ear (Raft et. Al., Dev Bio 2014), ephrin-B2 would be more important for the hearing function than ephrin-As. On the other hand, the functional ligand of EphA2 in the lens is thought to be ephrin-A5 (Cooper et. Al., PNAS 2008). We discussed these points in the revised manuscript (p18 line 19 to page 19 line 4).

Reviewer: Chattaraja et al., 2017 reported that in M1 subjects (one coding mutation of SLC26A4), the second allele in trans is a haplotype of 12 uncommon variants upstream of SLC26A in two independent cohorts, one from the United States and a second from Denmark. The EVA-associated haplotype (called CEVA) is common but apparently not in Asian populations. This study was not mentioned in Li et al.,. Do the two of 40 EVA subjects with one variant of EPHA2 and one coding variant of SLC26A4 have the CEVA haplotype? In the Chattaraja et al., 2017, CEVA appears to be the missing variant in most of their M1 EVA subjects. CEVA was reported to be rare in Asian populations but may not be entirely absent. 2 of 41?

Reply: Thank you for pointing out an important issue. CEVA is characterized by EVA without goitre (Chattaraja et. Al., J Med Genet 2017). While two Japanese patients exhibit goiter, we confirmed that those subjects do not have the CEVA haplotype by sequencing. We discussed this point on page 13 line 22 to page 14 line 2.

Reviewer: On page 16, there is no statement that all of the 41 human subjects in the study by Li and co-authors were ascertained after approval by a Helsinki committee or IRB and that the subjects signed an informed consent for participation in the study and the use of their genomic DNA. Was the study of human subjects in Li et al. formally approved by an institution where the authors are located? Please add this information. If not, discuss this issue

with the editors of Nature Communications who will consider how this might impact publication.

Reply: Thank you for pointing out the ethic issue in this paper. Our study is approval by IRB. Thus, we include this information on page 20 line 20 to p21 line 2.

Minor issues are described below first for the, title, figures and then in order of appearance in the text.

Reviewer: The title of the manuscript “Epha2 is an additional causative gene of Pendred syndrome with pendrin” is strange. “... Pendred syndrome with pendrin” is ambiguous. In the title, there is a gene name (Epha2), then the name of a syndrome (Pendred) followed by the name of a protein (pendrin). Think up a better title for your manuscript. You may well have an example of digenic inheritance of EVA.

Reply: Thank you for your suggestion. The title of our revised manuscript is to “Digenic inheritance of mutations in *EPHA2* and *SLC26A4* in pendred syndrome”.

Reviewer: Figure 1, panel b, what is the column labeled “Pass”? Nothing mentioned in the legend about “Pass”. Add size standards to images in panels a and b. What is your interpretation of all the bands in panel b, second column?

Reply: Thank you for your comment to improve our manuscript. Accordingly, we included requested information in Fig. 1a, b and Figure legend. Among the bands in panel b second column, the band corresponding to Epha2 molecular weight is indicated. Other bands seem to be binding partners of Epha2. Although very sensitive, silver staining gives semi-quantitative rather than quantitative results. The linear range for plots of stain density versus ng protein varies from protein to protein with different saturation levels.

Reviewer: Figure 2, panel b, the stria is thinner in the Epha2 ko. Can you tell which of the three cell layers (marginal, intermediate, basal) of the stria contribute to the thinning quantified nicely in panel c? In Figure 3, KCNJ10 staining is reduced. In what cell layer(s) of the stria vascularis is KCNJ10 known to be located? Since the stria is thinner compared to the wild type, is the endocochlear potential (EP) also reduced?

Reply: After re-examine stria vascularis of our Epha2 knock out mice, we found intermediate cells contribute most to the reduce of thickness (Fig. 2b, c). it is also where KCNJ10 is expressed.

Unfortunately, we could not measure endocochlear potential in mice inner ear because of the technical limitation in our institute. Since KCNJ10 is a very important potassium channel, we assume EP in Epha2 KO mice inner ear were affected.

Reviewer: Figure 5, the variants of SLC26A4 were reported by others and should be referenced. In the legend, add a “the” between “from” and Plasma membrane” Also, “Some disease forms of pendrin...” Do you mean by “forms”, mutant alleles/pathogenic variants?

Reply: Thank you for your comments. We corrected our description in the figure legend for Fig.5 and Fig. S4.

Reviewer: Figure 6, the arrow in the CAT scan points to an enlarged vestibular aqueduct. Mention this in the legend.

Reply: Accordingly, we add requested description.

Reviewer: Results, page 6, add “an” between “.... carried out..” and “...interactome analysis.”

Reply: Accordingly, we corrected description.

Reviewer: Page 13, delete “to” after “... membrane is not...”

Reply: We appreciate how carefully and thoughtfully you examine each piece of data in this paper. The related technical problems were already fixed in our new manuscript.

Reviewer #3:

Reviewer: The evidence for direct binding of pendrin to EphA2 in vivo is not very strong. Co-IP expts as in Fig. 1d are not well controlled. Since the authors have access to EphA2^{-/-} mice, similar Co-IP expts should be done with EphA2^{-/-} tissue lysates to exclude the possibility that pendrin is brought down with anti-EphA2 Abs unspecifically.

Reply: We highly appreciate your comments to improve the quality of our manuscript. Accordingly, we carried out immunoprecipitation with EphA2 KO mice tissue. New data are shown in Fig. 1d. We confirmed that pendrin was specifically co-immunoprecipitated with EphA2 using the EphA2 antibody.

Reviewer: EphA2^{-/-} mice have inner ear abnormalities. I find the morphological analysis unsatisfying. In Fig. 2a, the size of the cochlear duct seems “slightly enlarged”. Such a statement needs to be supported with quantified data. In Fig. 2b, the thickness of the stria vascularis is reduced. It is unclear from the images how and where the thickness was measured. As a minor point, in Fig. 2c, the y axis should always start at 0 um. In Fig. 2e, the number of mice for the quantification was not indicated.

Reply: First of all, to show better images of the inner ear, we performed micro-CT scans. Consistent with our previous observation with the light microscopy, cochlear lumen in EphA2 KO mice was significantly increased (Fig. 2a, b).

In line with reviewer #2's comment, we re-analyzed the stria vascularis, which is consisted of three layers of cells; marginal, intermediate, and basal cells. Especially, the cell layer of intermediate cells was reduced (Fig. 2c-e)

Reviewer: In Fig. 3, pendrin mislocalization in EphA2^{-/-} mice is fairly convincing. However, in Fig. 3d, the reduction of KCNJ10 immunoreactivity in EphA2^{-/-} mice should be quantified in groups of n=3 mice.

Reply: Accordingly, we quantified signals corresponding KCNJ10 and added in the revised manuscript (Fig. 3d, e).

Reviewer: The authors claim that ephrinB-B2 is an atypical ligand of EphA2. Since this is against current models of ephrin/Eph interactions, the authors need to provide better support

for this claim. Biophysical interaction studies, e.g. using Biocore, are needed in comparison with EphA4. In Fig. 4b,c, the Co-IP expts are not well controlled. Anti-ephrin-B2 IPs may bring down many proteins unspecifically. In Fig. 5e, the rate and timing of EphA2 internalization should be quantified. Similar expts in Fig. 5d look very different. While the rate of EphA2 internalization seems rather different between ephrinB-B2 and A1, the rate of pendrin internalization looks exactly the same (although again not quantified). The conclusion that both events are causally linked requires more experimental support.

Reply: Thank you for your comment to improve the quality of our manuscript.

Accordingly, we calculated the K_d value between the ephrin-B2 and the EphA2. The fusion protein of the EphA2 extracellular domain was immobilized onto the ELISA plate and binding of that of ephrin-B2 was examined. The K_d value between ephrin-B2 and EphA2 was 85nM, while that between EphA2 and ephrin-A5, the classic ligand of EphA2 was 2nM. The binding of ephrin-B2 with EphA4 as a ligand has been analyzed. Especially, a previous observation by Bowden et al. (2009, Structure volume 17, 1386-1397), has determined the K_d values of EphA4/ephrin-B2, EphA4/ephrin-A4 and EphA4/ephrin-A5 respectively. (EphA4/ephrin-B2, 10.8μM; EphA4/ephrin-A4, 360nM; EphA4/ephrin-A5, 36nM). The K_d value of EphA4/ephrin-B2 was approximately 30 times greater than that of EphA4/ephrin-A5 and 300 times greater than that of EphA4/ephrin-A4. Thus, the K_d value of EphA2/ephrin-B2 was within a reasonable range.

Reply: In addition, we examined immunoprecipitation of ephrin-B2 using control and EphA2 KO mice kidney lysate. EphA2 was co-immunoprecipitated with the ephrin-B2 antibody from control mice but not from EphA2 KO mice tissue (Fig. 4b).

Accordingly, requested blots were quantified. The results were presented in the revised manuscript (Fig. 4).

To further support our conclusion, EphA2 was knocked-down in the cultured cells. Consistent with EphA2 KO mice observations, loss of EphA2 in cultured cell resulted in reduced amount of pendrin on cell surface (Fig. S3c, d). Importantly, cell surface pendrin in EphA2 KD was not internalized after ephrin-B2 stimulation, suggesting that pendrin internalization via ephrin-B2 is in a manner dependent on EphA2 (Fig. S3c, d).

Moreover, we re-examined EphA2 auto-phosphorylation by ephrin-B2. Stimulation of EphA2 with ephrin-B2 Fc induced auto-phosphorylation of EphA2 in 30 minutes, which was sustained at least up to 2 hours. Of note, pendrin internalization triggered by ephrin-A/B stimulation was observed in 2 hours after stimulation. The level of EphA2 auto-phosphorylation induced by ephrin-B2 was weaker than that by ephrin-A1, however, treatment of human Fc with the cells failed to induce EphA2 autophosphorylation (Fig. 4d, e).

After stimulation with ligands, receptor tyrosine kinases (RTKs) are internalized and transported to early endosome. Consistently, we observed co-localized immunoreactive signal corresponding to ephrin-B2, phospho-EphA2 and EEA1, a marker of early endosome, 2 hours after ephrin-B2 stimulation (Fig. S3e, f).

As pointed out by the reviewer, the kinetics of EphA2 internalization induced by ephrin-B2 and ephrin-A1 is different. On the hand, that of pendrin internalization does not show a difference (Fig. 4f-h). This might be due to the stoichiometry of the protein complex between pendrin and EphA2. However, these results indicate that ephrin-B2 is the functional ligand of EphA2 to control it's membrane localization.

Reviewer: In Fig. 5, the regulation of pendrin localization by EphA2 is preliminary. It is unclear to me why disease forms of pendrin were co-IPed with a truncated version of EphA2 consisting only of the cytoplasmic and transmembrane domains. First of all, interactions of pendrin mutants with full-length EphA2 should be investigated, and the data should be quantified across several blots. Second, cytosolic mutants of pendrin that bind to EphA2 should be further analyzed, e.g. using immunofluorescence imaging. Co-expression with EphA2 should recruit (at least some of) those mutants to the plasma membrane, and ephrin stimulation should exclude them again from the PM. In Fig. 5c, it is unclear what is depicted in left and right panels. In Fig. 5d, the rates of pendrin and EphA2 internalization should be quantified.

Reply: Accordingly, we performed immunoprecipitation of full length of EphA2 with disease forms of pendrin and confirmed our previous results using a truncated version of EphA2. The results were presented with quantification in the revised manuscript (Fig. 5a, b).

To gain further insight into the role of EphA2 on pendrin regulation, pendrin A372V, L445W, Q446R or G672E was co-overexpressed with EphA2. The cells were transfected with cDNAs of encoding myc-pendrin disease forms with that of EphA2, and the non-permeable cells were stained with an anti-myc antibody. While signal corresponding to myc-pendrin was observed in approximately 65 % of cells, the ratio of myc-pendrin A372V, L445W, Q446R, or G672E positive cells was significantly decreased (Fig. S5a, b). Under these conditions, co-expression of EphA2 did not affect protein expression level of these disease forms of pendrin (Fig. 5a) but partially restored membrane localization of myc-pendrin A372V, L445W or Q446R (Fig. 5a, b). On the other hand, EphA2 overexpression did not affect localization of G672E.

Accordingly, Fig. 5c and Fig. 5d are fixed in the revised manuscript.

Reviewer: In Fig. 6d, the patient CT scan is not useful unless a similar scan from a healthy person is shown.

Reply: Accordingly, a CT scan of healthy person is provided in the revised manuscript (Fig. 6d).

Reviewer: In Fig. 7, the analysis of EphA2 mutations is incomplete and confusing. The biggest problem is that these EphA2 mutations do not affect EphA2's ability to bind wild-type pendrin (Fig. 7e). Hence, the mechanism by which these mutations affect inner ear function remains unclear. Perhaps, the association between EphA2 and pendrin is weaker? The authors did not perform expts as in Fig. 4e (pendrin internalization after ephrinB stimulation). Moreover, the blots shown in Fig. 7c,d should be quantified.

Reply: Accordingly, the effect of EphA2 mutations on pendrin internalization is examined. When EphA2 T511M or G355R are stimulated with ephrin-B2, internalization of pendrin as well as EphA2 is compromised (Fig. 7c). Pendrin S166N is known to have normal channel activity, but we showed that its internalization by ephrin-B2 was compromised (Fig. 5e, f). Given the phenotype of ephrin-B2 inner ear epithelial specific KO mice (Raft et al 2014 Dev Bio), ephrin-B2 is indispensable in the inner ear. These observations suggest the loss of binding ability of EphA2 to ephrin-B2 could cause inner ear dysfunction. Generally, the channel or transporter protein function, which is often controlled by their plasma membrane

localization is very important for tissue homeostasis. Both loss of function (eg. CFTR) and gain of function (eg. KCNQ1) can cause human disease. Proper membrane localization of pendrin might be important for its function. All of the blots in Fig. 7 are quantified and presented in the revised manuscript.

Reviewers' comments:

Reviewer #1 (Remarks to the Author):

Malformation of the inner ear is a characteristic of pendred syndrome, for which the chloride transporter pendrin was identified as the causative gene. Some mutations in pendrin affect its cellular location and/or trafficking of the protein. Interestingly, mutations in ephrinB2, a ligand for Eph receptor tyrosine kinases, also shows abnormal development of the inner ear. The authors here identify the EphA2 receptor as a molecule as another causative gene for Pendred syndrome.

The authors show an interaction between these proteins, and extensively characterize the changes in subcellular localisation of pendrin in ephA2 knockout mice. Data presented suggest that EphA2 is a molecule facilitating pendrin localisation to the (apical) membrane. While ephrinB2 is not a bona fide ligand for EphA2, the authors show that there is indeed some specificity in binding between EphA2 and ephrinB2, giving further meaning to the finding of inner ear defects in ephrinB2 mutant mice. Interestingly, some of the patients with pendred syndrome have mutations in EphA2 which are in the presumptive interaction area with ephrinB2.

The study is very well performed, and their conclusions well documented by their data, though some of the figures would profit from a better documentation, for example the naming of the proteins used for IP.

I would like to see the expression pattern of ephrinB2 protein in relation to EphA2 protein.

Is there any information when during development these defects emerge?

Is there any co-localisation of EphA2 protein with ephrinA ligand proteins?

Do ephrinA ligand mutants have defects in inner ear development?

Figure 1: the relative concentrations of pass-through and precipitated EphA2 appear inappropriate with 20% of input versus 80% of input, since the precipitate appears at least 10-20x fold stronger than the pass.

Reviewer #2 (Remarks to the Author):

The response to reviewers' comments and suggestions area acceptable. But there are some minor issues that need to be resolved and the abstract is difficult to understand.

1. Abstract, second sentence, Considering adding "appear to carry" so that the sentence would read--. "While biallelic mutations of the SLC26A4 gene, encoding pendrin, cause nonsyndromic hearing loss with EVA or Pendred syndrome, a considerable number of patients appear to carry a mono-allelic mutation." This conclusion depends on how

comprehensively a gene was screened for variants. Genes are much more complicated than exons, UTRs and splice junctions. It is likely that the authors did not eliminate the possibility of a pathogenic regulatory variant of SLC26A4 as they largely have not been characterized for this gene. The authors did check the published noncoding CEVA variants associated with EVA and didn't find any of them. These data were a nice addition to the revised manuscript. Nevertheless, the authors are urged to be careful not to rule out a regulatory variant of SLC26A4 as the missing second allele and that their hypothesis about digenic inheritance of EVA in human involving EPHA2 might not be correct despite formidable supporting data from the mouse in the manuscript. Leave some room for a mistaken conclusion. That's good science.

2. Abstract, line 77-78. "with weakly inducing EphA2 autophosphorylation, ..." is confusing wording.

3. Abstract, line 72, Do variants of EPHA2 by themselves in humans "cause Pendred syndrome? If there is digenic inheritance with a variant of SLC26A4, then the statements on line 72 and line 131 in the Introduction are not correct and probably not what the authors really intended to say.

Minor edits

Line 177, Interestingly, a specific...

Line 420. Add "an". "... suggesting an indispensable role..."

Line 467, "...Marshall Syndrome" not "Syndrom"

Reviewer #3 (Remarks to the Author):

The authors have answered all my questions with new results which support their main conclusions. Quantification of biochemical and imaging data were done appropriately.

We would like to thank the reviewers for their time and helpful comments. Please find below a detailed point-by-point response to all remaining questions.

Reviewer #1

Reviewer: The study is very well performed, and their conclusions well documented by their data, though some of the figures would profit from a better documentation, for example the naming of the proteins used for IP.

Answer: We are very grateful for this positive assessment. Accordingly the naming of the proteins used for IP was improved in Figure 1d and Figure 4b.

Reviewer: I would like to see the expression pattern of ephrinB2 protein in relation to EphA2 protein.

Answer: According to the reviewer's suggestion, the immunostaining images with an anti-ephrin-B2 antibody in the inner ear is shown in Sup Fig 3g. The signal corresponding to the anti-ephrin-B2 antibody was colocalized with that to the anti-EphA2 antibody (Sup Fig 3g). This ephrin-B2 immunostaining was consistent with previous report showing ephrin-B2 expression with the LacZ reporter mice (Bianchi et al., *Jhc* 2002).

Reviewer: Is there any information when during development these defects emerge?

Answer: Pendred syndrome is the congenital hearing loss, suggesting the importance of the developmental process of the inner ear. Consistently, pendrin KO mice exhibit inner ear malformation at E15.5 (Everett et al., *Hum Mol Gen* 2001). In addition, the gene deletion of ephrin-B2 in the inner ear results in malformation at E15.5 (Raft et al., *Dev Biol* 2014). Given the mechanistical insights into pendrin localization regulated by EphA2, it is likely that EphA2 KO mice may have the inner ear defects at E15.5. This is discussed at page 17 line 20-.

Reviewer: Is there any co-localisation of EphA2 protein with ephrinA ligand proteins? Do ephrinA ligand mutants have defects in inner ear development?

Answer: While no previous study has shown immunostaining of ephrin-As in the inner ear spiral prominence cells, it has been reported that at least ephrin-A3, ephrin-A4 and ephrin-A5 are expressed in the cochlea (Pickles, 2003 *Hearing Research*, Saeger et al., *Dev Dyn* 2011). Regarding mice phenotype, function of ephrin-A5 in the inner ear was investigated (Defourny et al., 2013 *Nature Comm*). However, no reports have shown that any ephrin-A mutants exhibit inner ear malformation. This would be due to the functional redundancy of ephrin-As. On the other hand, inner ear epithelial cell specific ephrin-B2 KO mice are known to show inner ear malformation, indicating an indispensable role of ephrin-B2 in the inner ear. We have identified EphA2 mutations from Pendred syndrome patients carrying mono-allelic mutation on the *SLC26A4* gene. These mutations affect the binding ability of EphA2 to ephrin-B but not ephrin-A, further supporting the importance of our findings. Please see page 16 line 22.

Reviewer: Figure 1: the relative concentrations of pass-through and precipitated EphA2 appear inappropriate with 20% of input versus 80% of input, since the precipitate appears at least 10-20x fold stronger than the pass.

Answer: This is due to the different amount of charge between pass-through and immunoprecipitated protein for western blotting analysis. While 1.5% of total pass-through was used for western blotting analysis, 7.5% of immunoprecipitated protein was charged. To avoid misunderstanding, this information was included in the figure 1a.

Reviewer #2

Reviewer: The response to reviewers' comments and suggestions area acceptable. But there are some minor issues that need to be resolved and the abstract is difficult to understand.

Answer: Thank you for high evaluation and useful comment to improve our revised manuscript. Accordingly, the abstract is modified.

Reviewer: 1. Abstract, second sentence, Considering adding “appear to carry” so that the sentence would read--. "While biallelic mutations of the SLC26A4 gene, encoding pendrin, cause nonsyndromic hearing loss with EVA or Pendred syndrome, a considerable number of patients appear to carry a mono-allelic mutation." This conclusion depends on how comprehensively a gene was screened for variants. Genes are much more complicated then exons, UTRs and splice junctions. It is likely that the authors did not eliminate the possibility of a pathogenic regulatory variant of SLC26A4 as they largely have not been characterized for this gene. The authors did check the published noncoding CEVA variants associated with EVA and didn't find any of them. These data were a nice addition to the revised manuscript. Nevertheless, the authors are urged to be careful not to rule out a regulatory variant of SLC26A4 as the missing second allele and that their hypothesis about digenic inheritance of EVA in human involving EPHA2 might not be correct despite formidable supporting data from the mouse in the manuscript. Leave some room for a mistaken conclusion. That's good science.

Answer: Thank you for pointing out important issue. Accordingly, we modified the sentence in the abstract.

Reviewer: 2. Abstract, line 77-78. “with weakly inducing EphA2 autophosphorylation, ...” is confusing wording.

Answer: Thank you for pointing out an important issue. Accordingly, the description was corrected.

Reviewer:3. Abstract, line 72, Do variants of EPHA2 by themselves in humans “cause Pendred syndrome? If there is digenic inheritance with a variant of SLC26A4, then the statements on line 72 and line 131 in the Introduction are not correct and probably not what the authors really intended to say.

Answer: Thank you for pointing out an important issue. We corrected the sentence in the abstract “ Here we identify *EPHA2* ass another causative gene of Pendred syndrome with *SLC26A4*.”

Reviewer: Minor edits

Line177, Interestingly, a specific...

Line 420. Add “an”. “,... suggesting an indispensable role...”

Line 467, “...Marshall Syndrome” not “Syndrom”

Answer: Thank you for your careful reading to improve quality of the manuscript. As pointed out, we corrected our manuscript.

Reviewer #1 (Remarks to the Author):

The response to my comments and suggestions are appropriate, and help clarify unclear issues with the manuscript.